# Ingression-type cell migration drives vegetal endoderm internalisation in the *Xenopus* gastrula

**Jason WH Wen, Rudolf Winklbauer***

Department of Cell and Systems Biology, University of Toronto, Toronto, Canada

**Abstract** During amphibian gastrulation, presumptive endoderm is internalised as part of vegetal rotation, a large-scale movement that encompasses the whole vegetal half of the embryo. It has been considered a gastrulation process unique to amphibians, but we show that at the cell level, endoderm internalisation exhibits characteristics reminiscent of bottle cell formation and ingression, known mechanisms of germ layer internalisation. During ingression proper, cells leave a single-layered epithelium. In vegetal rotation, the process occurs in a multilayered cell mass; we refer to it as ingression-type cell migration. Endoderm cells move by amoeboid shape changes, but in contrast to other instances of amoeboid migration, trailing edge retraction involves ephrinB1-dependent macropinocytosis and *trans*-endocytosis. Moreover, although cells are separated by wide gaps, they are connected by filiform protrusions, and their migration depends on C-cadherin and the matrix protein fibronectin. Cells move in the same direction but at different velocities, to rearrange by differential migration.
DOI: https://doi.org/10.7554/eLife.27190.001

## Introduction

The basic body plan of metazoans is established by gastrulation, and at its core is the movement of endoderm and mesoderm from the surface to the interior of the embryo. Among invertebrates, the pre-gastrulation embryo typically consists of a single-layered epithelium, and a common mechanism of germ layer internalisation is invagination, the inward bending of an epithelium at a pre-localised site. A classic example of gastrulation by invagination is the sea urchin embryo (*Kominami and Takata, 2004*), and more recently, the invagination of the mesoderm during gastrulation in the fruit fly *Drosophila melanogaster* has been thoroughly studied (*Rauzi et al., 2013*). Another major internalisation mechanism is ingression, where individual cells leave the epithelial layer to move interiorly. Both modes of internalisation can occur in the same organism. For example, primary mesenchyme ingression precedes invagination in the sea urchin embryo (*Katow and Solursh, 1980*; *Kominami and Takata, 2004*).

Within chordates, cephalochordates and tunicates develop from a single-layered blastula. Ingression is not observed in these groups, and internalisation of germ layers occurs by invagination (*Shook and Keller, 2008*). Although the blastula wall is single-layered in ascidian tunicates, it is thick relative to the size of the embryo, and the vegetal cells in particular are comparatively large, which gives ascidian invagination a distinctive appearance (*Satoh, 1978*; *Sherrard et al., 2010*). The transition to the third chordate group, vertebrates, is characterised by a sharp increase in egg size along with the formation of a thick multilayered epithelium that surrounds a blastocoel cavity. Whereas the animal side of the embryo can secondarily become single-layered, the vegetal half always remains as a multilayered cell mass. The corresponding ancestral mode of vertebrate gastrulation, conserved in lampreys, lungfish, and amphibians (*Collazo et al., 1994*; *Shook and Keller, 2008*), must adapt to this condition. In a second wave of further egg size increase, meroblastic cleavage again requires

***For correspondence:**
r.winklbauer@utoronto.ca

**Competing interests:** The authors declare that no competing interests exist.

**eLife digest** In most animals, the early embryo consists of a single layer of cells that forms a hollow sphere. This simple structure first gains complexity by organising into multiple layers that are fated to become specialised tissues in the adult, such as muscle or skin. To form the primitive gut, a group of cells known as the endoderm must move from the surface to the interior in a process called gastrulation.

Since the early 1900s, the sea urchin and the frog have been the standard species used to study gastrulation. In sea urchin embryos, gastrulation entails bending the sheet of cells that form the surface of the embryo inward at a predetermined site to generate a pocket that will become the digestive system. By contrast, frog embryos begin gastrulation as multilayered structures, and the embryo's surface does not bend. Furthermore, classic studies of frog gastrulation have found that cells do not leave the surface of the embryo to enter its interior. So despite generations of students having learned about how gastrulation occurs in backboned animals from studying frogs, the cell behaviours that internalise the endoderm are still not understood.

Wen and Winklbauer now show that endoderm cells in the frog move using the same set of behaviours that cells in other organisms use to break loose from or bend sheets of cells. Individual cells move by simultaneously pushing their front end forward while retracting their rear in a peculiar manner, by engulfing their own cell surface at a large scale. In the frog embryo, this movement is coordinated into an organised pattern where cells use the surfaces of their slower or stationary neighbours to propel past each other, and then slow down to return the favour. This constitutes a newly defined type of movement referred to as ingression-type migration.

Frog embryos are remarkably large because each of their cells is packed with yolk to support development until the animals are able to feed. As an adaptation to this large size, some frog gastrulation movements appear unusual. However, Wen and Winklbauer show that the cell behaviours that drive these movements are similar to the behaviours of cells in single-layered embryos, and indeed the behaviour of single-celled organisms such as amoebae. Further research is now needed to investigate how these cells find their way straight to the interior of the embryo.
DOI: https://doi.org/10.7554/eLife.27190.002

adaptation of gastrulation movements in various vertebrate groups. For example, germ layer internalisation occurs by ingression at a novel structure, the primitive streak, in birds and mammals (*Arendt and Nübler-Jung, 1999*).

In the ancestral mode of vertebrate gastrulation, mesoderm is internalised by involution or ingression at the blastopore lip, and the supra-blastoporal endoderm by involution (*Shook and Keller, 2008*). The multilayered structure of the sub-blastoporal endoderm of the vegetal cell mass precludes invagination, and ingression of the vegetal surface is also absent. Thus, the question arises of how the vegetal endoderm is internalised. Surprisingly, despite endoderm internalisation being a defining feature of gastrulation, it has scarcely been studied in lower vertebrates. Even in the African clawed frog, *Xenopus laevis*, the most extensively studied model of vertebrate gastrulation, the inward movement of the sub-blastoporal vegetal endoderm has only been analysed at the tissue level (*Winklbauer and Schürfeld, 1999*; *Ibrahim and Winklbauer, 2001*; *Papan et al., 2007*).

In *X. laevis*, the cone-shaped vegetal endoderm is initially narrow inside at the blastocoel floor (BCF), and wide at its outer, epithelial surface. At the equator, it is surrounded by an annulus of mesoderm (*Figure 1A*) (*Keller, 1975*; *Keller, 1976*; *Bauer et al., 1994*). At the onset of gastrulation, the vegetal cell mass surges animally into the embryo. It narrows at its vegetal-most part, expands at the BCF, rolls the anterior mesoderm against the ectoderm and displaces the posterior mesoderm in the vegetal direction (*Figure 1A*) (*Winklbauer and Schürfeld, 1999*; *Ibrahim and Winklbauer, 2001*; *Papan et al., 2007*; *Winklbauer and Damm, 2012*). When a mid-sagittal slice of the vegetal half of the gastrula is explanted (*Figure 1B*), the entire process continues in isolation and appears as rotational movements on the dorsal and the ventral sides, which gave rise to the term vegetal rotation (*Video 1*). Further dissection of explants revealed that vegetal rotation is based on active, region-specific tissue deformations within the vegetal cell mass (*Winklbauer and Schürfeld, 1999*; *Ibrahim and Winklbauer, 2001*). However, the cellular mechanisms that drive vegetal rotation are

**Figure 1.** Tissue movements during *Xenopus laevis* gastrulation. (**A**) Fate map and tissue deformation of *X. laevis* germ layers for stages 10–13. Movements of the ectoderm (white), mesoderm (blue), and endoderm (yellow) are indicated (top row). Blastocoel floor expansion throughout developmental stages is shown (red line). Mid-sagittally fractured gastrulae at stages 10–13 (mid row). Animal is to the top, vegetal to the bottom, ventral to the left, and dorsal to the right. Early, mid, and late stage gastrulae are shown together with the corresponding developmental stage and timeline (bottom row). The onset of gastrulation is set as 0:00 in hours and minutes. Blastocoel (bc) and archenteron (arc) are indicated. (**B**) Schematic of vegetal explant. The ectodermal BCR was removed with incisions shown (red lines). A mid-sagittal slice of about 5 cell layers thick was removed from the vegetal half of stage 10 embryos and placed under a coverslip for observation. Discarded regions are indicated (X's). Arrows indicate that the explant was tilted 90° toward the viewer to provide an overhead view, and then flipped back to the sagittal view.
DOI: https://doi.org/10.7554/eLife.27190.003

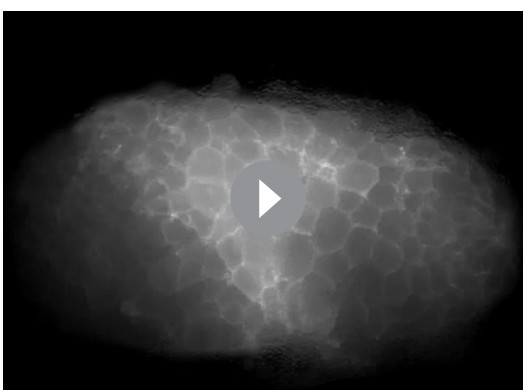

**Video 1.** Vegetal 'slice' explant. Explant was excised from a stage 10 gastrula. Movie shows tissue autonomous vegetal rotation movement over the course of 60 min. Cells are labelled with membrane-GFP. Animal is to the top, vegetal to the bottom.
DOI: https://doi.org/10.7554/eLife.27190.004

not known, and it is not understood how these processes are related to other modes of gastrulation in chordates.

In the present work, we have analysed the cellular mechanism of vegetal rotation. We show that endoderm cells undergo elongation and region-specific re-orientation at the onset of gastrulation and move by amoeboid migration without involving lamellipodial, filopodial or bleb-like protrusions. Spatially graded differences in movement velocity lead to orderly cell rearrangements as cells move over and between each other. Rearrangement by such differential migration narrows the vegetal-most part of the tissue and expands the animal part, which leads to the inward surge of the vegetal mass. During migration, endoderm cells are separated by wide interstitial gaps, which are bridged by dynamic filiform protrusions. Despite these gaps, C-cadherin is required to maintain cell migration, and interaction with the extracellular matrix (ECM) protein fibronectin (FN) is also necessary. A peculiar mode of ephrinB1-dependent trailing edge retraction by macropinocytosis and *trans*-endocytosis, combined with the amoeboid characteristics of endoderm cell translocation, suggests that vegetal rotation is a modification of invagination or ingression adapted to the multilayered structure of the vegetal mass, and is based on ingression-type cell migration.

## Results

### The BCF expands by endoderm cell rearrangement and oriented cell shape changes

To analyse vegetal rotation, we first quantified tissue shape changes performed by vegetal slice explants. Beginning at gastrulation, the upper region of explants expanded, and the BCF widened by 1.8-fold in two hours before reaching a plateau. Simultaneously, the equatorial waist of explants narrowed by 0.75-fold (*Figure 2A*). At the cell level, cell rearrangement within the exposed-surface plane of the explant, rearrangement by cells moving in and out of this plane, and cell elongation in region-specific patterns were apparent (*Figure 2B*). In the absence of strong cell division or cell growth during gastrulation (*Saka and Smith, 2001*; *Kurth, 2005*); four cells divided in explant shown in *Figure 2B*), changes in apparent cell size must be due to cell shape changes and incomplete intercalation in and out of the plane of view (*Figure 2B*). Together, the changes in cell shape and position expanded the upper part of the explant at the expense of the lower, narrowing part.

Quantitatively, overall cell rearrangement was indicated by changes in cell numbers along landmark lines over time. Average cell number increased from 11.7 to 17.3 at the BCF, remained unchanged along the animal–vegetal (A–V) axis, and decreased from 21.3 to 14.3 along the waists of explants (*Figure 2C*; *Figure 2—source data 1*). Cells also elongated slightly with the onset of rearrangement in explants (*Figure 2B*). In embryos, cell elongation was less pronounced, as seen by scanning electron microscopy (SEM) (*Figure 2E*). Cell elongation was accompanied by cell re-orientation and alignment, and elongation was predominantly in the direction of movement (*Figure 2D*; *Figure 2—figure supplement 1A*; *Figure 2—figure supplement 1A—source data 1*). After explant excision, cells were on average obliquely oriented in all regions. During the next half hour, cells in the top layers turned parallel to the BCF, while those in the middle layers aligned with the A–V axis. Thus, cells near the BCF became perpendicularly oriented relative to cells in the center. Orientation changed similarly in the embryo (*Figure 2D*; *Figure 2* Data; *Figure 2—figure supplement 1B—source data 1*). In particular, cells of the expanding BCF flattened in parallel to the tissue surface, perpendicular to cells located farther vegetally (*Figure 2E*). Cells of the vegetal epithelial layer are

Figure 2 image.

**Figure 2.** Cellular basis of vegetal rotation. (**A**) Tissue autonomous movement in live explants. Blastocoel floor (BCF) expansion was followed in explants (red line). BCF length was determined by tracking positions of peripheral endoderm cells (red dots). The equatorial waist (white dashed line) at the explant mid-point runs at the level of the dorsal blastopore (red arrowhead). Animal (An) is to the top, vegetal (Vg) to the bottom, ventral (V) to the left, and dorsal (D) to the right. (**B**) Cell behaviours in explants. Cells are outlined for the explant shown in *Figure 2A*, and morphogenetic cell

*Figure 2 continued on next page*

*Figure 2 continued*

behaviours from 30 to 90 min are indicated (coloured cells). Elongated marginal cells (purple zones) are shown with respect to the cells of the vegetal cell mass (white zone) of the equatorial waist (white and purple solid line). Cells that disappeared from the surface (Brown), cells that emerged at the surface from deep layers (Yellow), cells undergoing division (Blue), cells that reduced their area at the surface as they migrated into deep layers (Red), and cells that increased their surface as they spread out during migration (Green) are shown. (C) Quantification of cell numbers at the BCF and equatorial waist. Error bars indicate S.D. (D) Quantification of cell orientation. Rose diagrams indicate the number of cells in explants (left) or in embryos (right) oriented at given angles relative to the A–V axis in the top (yellow), mid (orange), and bottom (red) cell layers. At stage 10, the endoderm has an average of twelve cell layers, which were evenly divided into three regions. The lengths of bars indicate the number of occurrences in 5° bins. (E) Endoderm cells depicted in scanning electron micrographs of mid-sagittally fractured gastrulae. An apically constricted epithelial surface cell is indicated (asterisk). Corresponding stages are indicated on the left. A schematic of the region of interest (red box) is indicated in the top right corner of select panels. A ruler corresponding to the approximate position of top (yellow), mid (orange), and bottom (red) cell layers is shown in each panel. Panels show data from 14 embryos collected from different egg batches.

DOI: https://doi.org/10.7554/eLife.27190.005

The following source data and figure supplements are available for figure 2:

**Source data 1.** Quantification of cellular changes during vegetal rotation.

DOI: https://doi.org/10.7554/eLife.27190.007

**Figure supplement 1.** Quantification of cellular changes during vegetal rotation.

DOI: https://doi.org/10.7554/eLife.27190.006

**Figure supplement 1—source data 1.** Quantification of cellular changes during vegetal rotation.

DOI: https://doi.org/10.7554/eLife.27190.008

known to remain at the surface (*Keller, 1978*), but occasionally they showed apical constriction and became wedge-shaped (*Figure 2D*).

Our data support the notion that a combination of cell rearrangement and oriented cell elongation underlies the distinct shape change of vegetal explants. To estimate the relative importance of these processes, we considered in detail a representative explant (*Figure 2B*). Here, the narrowing of the equator was due to a decrease in cell number, as in other explants (*Figure 2C*). In part, this was because of a disappearance of cells at the lateral explant margins, which was to some extent offset by the elongation of former sub-marginal cells in parallel to the equator (*Figure 2B*). We suggest that this 'edge effect' was an explant artifact. In the remaining central section of the equator, cell numbers decreased by 0.64 (from 14 to 9 cells), matching the 0.64-fold decrease in equator length. Apparently, the slight elongation of cells perpendicular to the equator in the center of the explant was offset by a similarly slight net increase in apparent cell size. Cell disappearance was rare in the center of the explant (*Figure 2B*); thus, in-plane cell rearrangement constituted the major morphogenetic process to narrow the lower part of the explant.

Cells disappeared at the lateral margins both above and below the equator, and to a lesser extent sub-marginally below the equator. However, cells appeared at the explant surface only above the equator, which contributed to expansion of the explant upper region (*Figure 2B*). As shown below, this type of intercalation was mostly due to superficial cells moving laterally to expose deeper cells. Importantly, cell lengthening parallel to the BCF also contributed to the region's lateral expansion. Directly at the BCF, net cell number increased 1.3-fold (from 13 to 17 cells), and the remainder of the total 1.8-fold length increase (i.e. a 1.4-fold contribution) was due to oriented cell lengthening and some increase in apparent cell size (*Figure 2B*). As described in the following paragraph, cell lengthening is an integral part of endoderm cell movement; thus, cell rearrangement and its associated cell movements seemed to be the main mechanisms driving vegetal rotation.

## Amoeboid migration of endoderm cells

We next determined the mechanism by which vegetal cells rearrange. Generally, two basic processes of cell neighbour exchange have been identified. An intensively studied paradigm is epithelial cell intercalation by junction remodelling (*Bertet et al., 2004*). For example, in *D. melanogaster* gastrula ectoderm, a cell–cell boundary constricts and resolves to separate two neighbouring cells while a new, perpendicularly oriented contact is formed between previously non-attached cells. An analogous mechanism was proposed for mesenchymal cell rearrangement in *X. laevis* mesoderm (*Shindo and Wallingford, 2014*). However, mesenchymal rearrangement can also be driven by the migration of cells over each other. A defining feature of migration is that a cell establishes new

contacts on a substratum and detaches from previous contacts, thus changing its position. When two cells migrate over each other, one cell serves as substratum at a given instance for the other to translocate across it. For rearrangement by junction remodelling, no such distinction can be made as the common contact areas between two cells shrink or expand together.

During vegetal rotation, the endoderm cells rearranged by amoeboid migration (*Figure 3A*). While cells wedged themselves between neighbours, they underwent cycles of cell body elongation in the direction of movement, expansion of the cell front, narrowing of the cell rear, and retraction of the trailing edge. Cell shapes reminiscent of the amoeboid motility cycle were also seen in the embryo using SEM (*Figure 3B–C*). Whereas cell tails were often flattened against other cells, leading edges were blunt and locomotory protrusions such as lamellipodia or filopodia were notably absent.

To directly show that cells translocated relative to their neighbours, yolk platelets were used as markers of cytoplasmic positions (*Kubota, 1981*; *Selchow and Winklbauer, 1997*). Platelets within an advancing cell maintained their relative positions during migration (*Figure 3A*), which indicated that the cytoplasm of the cell body advanced as a whole. Cell displacement occurred relative to the yolk platelets of stationary cells on the sides, and contact with these cells was reduced at the rear while new contacts were formed at the front (*Figure 3A*). The leading edge of the cell remained in contact with the cell ahead of it, and both cells moved in tandem. On other occasions, cells invaded the space evacuated by the retraction of a cell (*Figure 4A*). At the rear, a lagging cell followed closely, although contact with that cell was gradually reduced (*Figure 3A*).

To confirm this mode of translocation, we followed the lengths of lateral contacts, and the distances of these from leading and trailing edges over a time interval (*Figure 3D*). While lateral contacts remained stationary, the leading edge advanced and the trailing edge retracted, which led to a net translocation of the cell relative to its neighbours. Again, the trailing edge reduced its contact with the cell behind, but membrane undulations suggesting cell detachment also occurred laterally. Occasionally, vesicles appeared inside the cell. Taken together, endoderm cell displacement shows the hallmarks of migration; that is, contact formation at the cell front and contact resolution at the rear. While lateral contacts remain stationary, the cytoplasmic content of a cell moves forward into the advancing front region.

## Endoderm cells rearrange by differential migration

During cell-on-cell migration, cells necessarily move past one another and therefore rearrange locally. To achieve tissue remodelling, rearrangements must be patterned globally such that small local cell displacements translate into large-scale shape change at the tissue level. In the endoderm, the more vegetal part narrows laterally, whereas the animal part expands. In the narrowing region, the elementary rearrangement event consisted of a merging of cell columns. In groups of contiguous cells, animal–vegetal neighbours separated because of the higher velocities of more animally positioned cells, whereas their lateral neighbours converged to fill the spaces left after separation (*Figure 4A*; *Video 2*). Cells also partially or completely disappeared into the deeper layers (*Figures 2B* and *4A*). To a lesser extent, this type of rearrangement also contributed to the narrowing of explants below the equator (*Figure 2B*).

For oriented rearrangements to occur across the whole expanse of the narrowing zone, cell velocity must increase continuously from vegetal to animal. Such a velocity gradient was observed (*Figure 4B*). Relative to the vegetal surface, cells moved faster when they were located farther animally, up to a zone near the BCF (*Figure 4D*; *Figure 4—source data 1*). Movement was also slightly faster in the center of the explant compared to the periphery (*Figure 4D*); that is, cell movements were not restricted to the endoderm periphery as previously suggested (*Winklbauer and Schürfeld, 1999*). The timing of explant excision could account for this discrepancy, as vegetal rotation initially spreads from the periphery to the center of the vegetal mass.

Rearrangement by the merging of cell columns is easily conceptualised for cells racing side by side on an external surface. However, difficulty arises in a multilayered tissue when a cell is supposed to move consistently slower than the cells directly ahead of it, while it must attach to them to propel itself forward. Using the front cells as substratum would imply that the rear cell moved faster than these cells and would overtake them. As a solution, we found that individual cell movements were intermittent and neighbouring cells usually translocated at different velocities (*Figure 4C*). Discontinuous movement allowed cells to temporarily use adjacent cells as substratum during bursts of locomotion, and in turn, these cells served as substratum for neighbouring cells in subsequent steps.

**A**

| cell body elongation | | leading edge expansion | trailing edge recession | trailing edge retraction |

mGFP

0min  3min  11min  19min  31min

An.
Vg.

Yolk Platelets

25μm

Δd  Δd  Δd  Δd

**B**

Embryo S.E.M.

An.
Vg.

1  3  1  2  2  3  4

50μm

**C**

Embryo S.E.M.

An.
Vg.

25μm

**D**

mRFP AvidinFITC

0min  12min  16min  18min  20min  26min  32min

mRFP

AvidinFITC

20μm

Illustration

**Figure 3.** Amoeboid cell behaviours. (**A**) Amoeboid migration of vegetal endoderm cells. mGFP-labelled cells (top row) are shown. Major cell shape changes (dashed arrows) are indicated. Yolk platelets in the same cells are shown in bottom row. A select cell (pink dashed outline) moves with respect to neighbouring cells (grey dashed outlines). Select yolk platelets (pink and yellow outlines) within the moving cell and platelets in a neighbour cell (blue and orange platelets) are indicated. Platelets in different cells move relative to each other, indicating cell migration. Degree of platelet

*Figure 3 continued on next page*

*Figure 3 continued*

displacement is indicated (Δd, white double arrow). (**B, C**) Endoderm cell morphology in the embryo, as seen in SEM. (**B**) Morphology is consistent with amoeboid movement, cells are numbered as showing (1) cell elongation, (2) leading edge expansion, (3) trailing edge recession, and (4) retraction. (**C**) Higher magnification of cells undergoing leading edge expansion (left), trailing edge recession (center), and trailing edge retraction (right). Cell front (yellow arrows) and rear (blue arrows) are indicated. (**D**) Coordinate behaviors during cell locomotion. An elongated cell maintains stable lateral borders (yellow arrowed line) throughout time interval (parallel dashed white lines). To advance, the leading edge is extended (green arrow) relative to its initial length (top grey line), while the trailing edge is retracted (red arrow) relative to its initial length (bottom grey line). Extracellular debris attached to the leading cell (blue arrow) and lagging cell (white arrow) are indicated to show displacement. Enlargement of the trailing edge shows contact reduction (mRFP panels, dashed line flanked by arrows) between cells. AvidinFITC puncta are visible (white arrows) at sites of membrane undulation. Interpretation of trailing edge retraction is depicted in bottom rows. AvidinFITC puncta are indicated (black arrows). Area in green corresponds to interstitial space. In all panels, animal (An) is to the top, and vegetal (Vg) to the bottom. Schematic showing the region of interest (red box) is indicated in the top right corner of select panels.

DOI: https://doi.org/10.7554/eLife.27190.009

Nevertheless, average velocity changed gradually in the direction of movement, which promoted cell rearrangement. The migration of single endoderm cells *in vitro* (see below) suggested that their translocation was intrinsically intermittent.

In the upper, laterally expanding zone, the A–V velocity gradient was inverted. In explants, cells that were at or near the BCF still moved fast, but changed the direction of their migration (*Video 3*) from animal to lateral (*Figure 4E,F*). Consequently, the A–V component of their velocity gradually diminished as they approached the BCF (*Figure 4E*). This reversal of the velocity gradient changed the direction of intercalation from mediolateral to animal–vegetal; faster cells coming from behind inserted themselves between the slower cells ahead. As cells did not slow down, but changed direction to move sideways, and because cells were elongated in the direction of their active movement, this cell reorientation also contributed to the expansion at the BCF (see *Figure 2B*). Lastly, as cells migrated away from the central axis, they intercalated not so much by merging of cell rows in the plane of the explant surface, but by exposing cells deeper in the explant; that is, by intercalation perpendicular to the surface plane (See *Figure 2B*). In the embryo, cells were also obliquely oriented below the BCF and parallel to the BCF at the surface (*Figure 4G,H*). Together, our data suggest that at the BCF, elongated cells move laterally and cells from deeper layers insert themselves into gaps that open, which expands the BCF. Eventually, the BCF cells will line the remnant of the blastocoel cavity, and as it shrinks and disappears, they will merge with the deep endodermal cell mass (*Ewald et al., 2004*).

Endoderm cells moved individually in the direction of their long axes (*Figure 3*), and because cells were not oriented strictly in parallel (*Figure 2D*), one would expect independent trajectories of adjacent cells. However, their mutual attachment integrated their diverging migration trends. As an expression of this effect, net cell movement and cell orientation were not fully aligned in explants. While adjacent cells move in parallel, their long axes diverged such that cells drifted sideways to some extent. Both active movement of the cells in the direction of their long axes and passive drift imposed by the surrounding cells appeared to contribute to their net translocation. When active and passive components are aligned, net movement should be fastest. This behaviour was indeed observed (*Figure 4—figure supplement 1*; *Figure 4—figure supplement 1—source data 1*), which indicates that cells do not migrate individually as within a rigid ECM scaffold, but mutually affect their trajectories.

Regional velocity differences were correlated with cell packing densities in the embryo (*Figure 4I, J*). Endoderm cells were close to each other near the vegetal pole, but became less densely packed toward the BCF. At a given A–V level, packing decreased toward the center of the embryo. The channels that separate cells were narrower near the vegetal base and wider more animally and centrally (*Figure 4K*; *Figure 4—source data 1*). Loose cell packing could reflect lower adhesion, and in calcium-free medium, explants do indeed dissociate faster in the animal region (Wen and Winklbauer, unpublished). In turn, low adhesion could facilitate migration through reduced resistance to movement. In summary, in both zones of the vegetal endoderm, rearrangement was based on gradients of cell velocity in the direction of movement toward the BCF, that is, on the average velocity increasing or decreasing in this direction. We refer to this migration-based intercalation mechanism as differential migration.

**Figure 4.** Differential cell migration. (**A**) Cell rearrangement in explants. A leading cell (blue) moves away from the lagging cell (red), neighbouring cells (orange, green) converge to fill the gap. Corresponding trajectories are shown (coloured arrows). (**B**) Cell displacement in explants. Displacements were recorded starting 15 min after explantation. Panels indicate changes in cell positions between 15–30, 30–45, and 45–60 min. Direction and magnitude of displacements (white arrows) are indicated. (**C**) Migration velocity variability in explants. Velocities correspond to displacements shown in (**B**). Cells 1–4

*Figure 4 continued on next page*

*Figure 4 continued*

move as described in (A). Colours refer to velocity scale (right). Grey cells were not tracked due to poor visibility. (D) Migration velocity in cell columns (left) and rows (right). Plots show average instantaneous velocities at different time intervals after explantation (colours). Bars indicate S.E. Schematic marks the explant center (Cn), periphery (Pr), animal (An), and vegetal (Vg) boundaries. Panels show data from three embryos from different egg batches. (E) Total velocity is maintained while its vertical component is reduced at the BCF as cells change orientation. Cell movement (top row). Cohort (top-mid row) shows a leading cell (green) advancing laterally relative to the lagging cell (orange), the gap that opens is filled by an inserting deep cell (purple). Cell near the surface (red) remained parallel to the BCF. A deep cell (blue) initially oriented perpendicular to the surface re-oriented to a parallel alignment with the BCF. Cell re-orientation (bottom-mid row) of deep cell (blue dashed outline) relative to its previous position (grey dashed outline) is indicated (arrow). Re-orienting cell velocity vectors (bottom) showing the vertical component (black arrow) relative to the total velocity vector (grey arrow). (F) Cell re-orientation. Optical section shows that the entire cell body is rotated during re-orientation. Movement is indicated (orange arrow). (G, H) Cell morphology in the embryo is consistent with cell reorientation and insertion at the BCF. SEM of cells at the BCF of stage 10.5 gastrulae (left), elongated cell is highlighted (orange, black arrow), blastocoel (bc) is indicated. endo, vegetal endoderm cells. (I, J) Interstitial gaps between cells in the embryo. (I) TEM section through the endoderm (left), negative of TEM (right). Gap width varies between the top (yellow arrow), mid (orange arrow), and bottom cell layers (red arrow). (J) A vertical series of gaps shows gap-width increase in a vegetal to animal direction. (K) Gap width in cell columns (left) and rows (right) in gastrulae. Error bars indicate S.E. Panels show data from six embryos from different egg batches. Schematic of the region of interest (red box) is indicated in the top right corner of select panels.

DOI: https://doi.org/10.7554/eLife.27190.010

The following source data and figure supplements are available for figure 4:

**Source data 1.** Quantification of differential cell migration.
DOI: https://doi.org/10.7554/eLife.27190.012
**Figure supplement 1.** Cell translocation is a result of active cell movement and passive drift imposed by surrounding cells.
DOI: https://doi.org/10.7554/eLife.27190.011
**Figure supplement 1—source data 1.** Quantification of cell alignment with respect to cell velocity.
DOI: https://doi.org/10.7554/eLife.27190.013

## Endoderm cells form contacts across large interstitial spaces

To investigate how cells could migrate across each other whilst separated by large gaps, we further characterised cell–cell interactions. When gastrulae were examined using SEM, wide interstitial spaces between endoderm cells were observed to be bridged by thin cell processes (*Figure 5A*). Under transmission electron microscopy (TEM), interstitial spaces 1 µm wide on average and containing ECM material were observed between laterally aligned cells; they widened into large gaps where several cells met (*Figure 5B*). Cells were directly attached to each other over short stretches only. In these close contact regions, membranes approached to within 30 nm (*Figure 5B–C*), a distance compatible with cadherin adhesion. Moreover, thin cytoplasmic protrusions formed stitch-like contacts between cells (*Figure 5B*). Thus, cells were mostly surrounded by an ECM-filled interstitial space, but were in direct contact in small areas, and through thin cell processes.

Gaps between cells were also seen in live explants (*Figure 5D*). In medium containing AvidinFITC, cells were separated by fluorescent spaces about 1 µm wide on average, similar to the gaps observed using SEM and TEM (*Figure 5E*; *Figure 5—source data 1*). In TEM, densely stained globular material on the surface of cells likely represented collapsed ECM

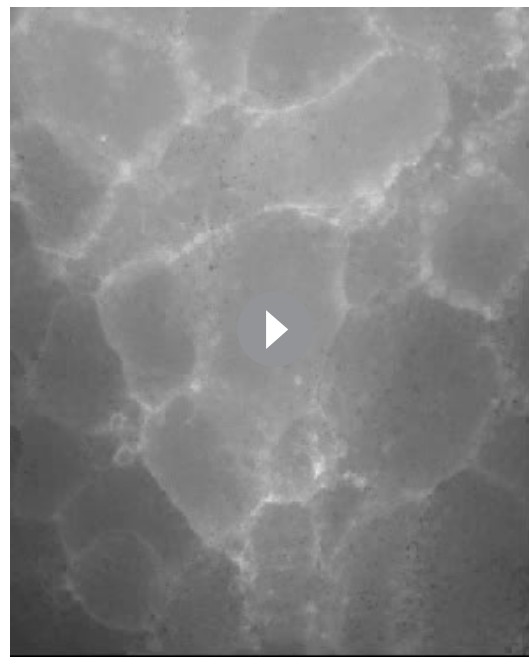

**Video 2.** Endoderm cell rearrangement. Neighbour exchange of central endoderm cells during differential migration viewed from a magnification of *Video 1*. Cells are labelled with membrane-GFP. Animal is to the top, vegetal to the bottom.
DOI: https://doi.org/10.7554/eLife.27190.014

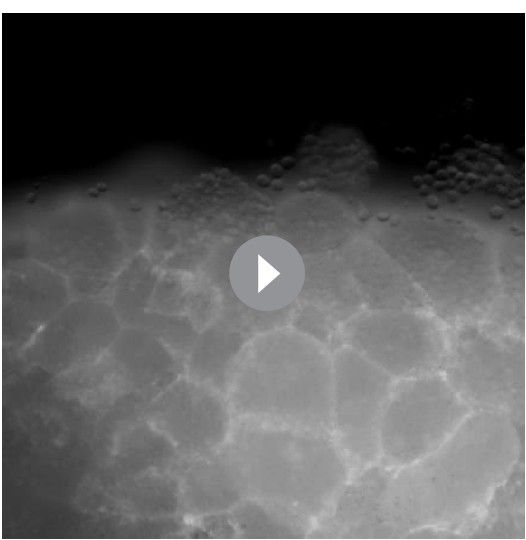

**Video 3.** Cell rearrangement at the BCF. Movement of cells near the BCF in explants viewed from a magnification of *Video 1*. Cells are labelled with membrane-GFP. Animal is to the top, vegetal to the bottom.

DOI: https://doi.org/10.7554/eLife.27190.015

material (*Figure 5F*). In live explants, long (1–10 μm), thin (0.5–1.0 μm), F-actin-filled protrusions were seen to connect cells for short time intervals while rapidly extending and retracting (*Figure 5D,G*). We observed that a 5 μm long protrusion could undergo a complete cycle of extension, attachment and retraction in under 3 min to provide highly dynamic, reversible cell contact. Amoeboid migration is commonly noted for its independence from specific substratum adhesion, but in the section below, we provide evidence to show that endoderm cells required defined molecular interactions with the ECM and with each other for proper cell migration.

## Fibronectin interaction is required for endoderm cell migration

A putative substratum molecule in the ECM is FN. Endoderm cells adhere strongly to the RGD cell binding site of FN *in vitro* (*Winklbauer, 1988*), and they secrete and accumulate FN on their surface (*Winklbauer, 1998*). However, in contrast to their *in situ* behaviour, isolated endoderm cells on FN adsorbed to tissue culture plastic are multipolar, extend lamellipodia, and yet do not migrate (*Wacker et al., 1998*; *Luu et al., 2008*). We confirmed that endoderm cells on FN/plastic were multipolar and extended numerous protrusions (*Figure 6A*). On gelatin-coated dishes, cells were non-adherent, but on a substratum of FN adsorbed to gelatin, cells adopted *in vivo*-like features (*Figure 6A–B*). They attached and elongated weakly, and established polarity through the development of discernable trailing ends, whereas lamellipodia were absent (*Figure 6A*), as in the embryo (*Figure 6B*).

We then examined cell migration on the different substrates (*Figure 6C–D*; *Video 4*). On FN/plastic, cells spread but did not translocate (*Figure 6C,E*). On gelatin alone, cells also remained stationary (*Figure 6E*; *Figure 6—source data 1*), but on FN/gelatin cells translocated, mimicking their migration in the tissue context (*Figure 6D*). Cells moved intermittently, at an average velocity of 4 μm/min, which was faster than in the intact tissue (*Figure 6E*). Thus, interaction with appropriately supplied FN is sufficient to support endoderm amoeboid migration. On FN/plastic or FN/gelatin, the onset of substratum-specific motile behaviour occurred within minutes, which indicates that the modulation of cell behaviour was unlikely due to changes in gene expression, but constituted a direct adaptation to the cell environment. Migration 'plasticity' in response to substratum properties is a well-documented phenomenon in many cell types (*Paluch et al., 2016*; *Te Boekhorst et al., 2016*).

To assess the requirement for FN *in situ*, we blocked cell–FN interaction using an Arg–Gly–Asp (RGD)-containing peptide that competitively inhibits integrin binding to FN. When injected into blastula stage embryos, RGD peptide caused cell rounding and detachment by the gastrula stage. Embryos injected with RGE control peptide appeared normal (*Figure 6F*). Thus, FN was required to maintain endoderm cell adhesion in the embryo. Exposure of vegetal slice explants to RGD peptide also perturbed cell morphology (*Figure 6G,H*; *Figure 6—source data 1*). Compared with control RGE peptide treatment, RGD-treated cells became significantly more rotund. However, because cells can deviate randomly from a spherical shape, we determined the component of elongation that paralleled the A–V axis, that is the approximate axis of migration (*Figure 6—figure supplement 1*). This 'elongation congruity' was unity in RGD-treated cells, which indicated that elongation was indeed random and not aligned with migration, in contrast to control peptide treated explants (*Figure 6I*; *Figure 6—source data 1*). Importantly, cells became nearly non-migratory upon RGD treatment, whereas RGE-treated cells moved at normal velocities (*Figure 6J*; *Figure 6—source data*

**Figure 5.** Vegetal endoderm cells migrate through wide interstitial spaces. (**A**) SEM of endoderm in embryos. Overview (left). High magnification (center) reveals interstitial spaces between cells (red arrows). Cells are linked by stitch contacts (right; yellow arrows). (**B**) TEM of endoderm in embryos. Overview (left), cell gaps (3- or 4 cell junctions; red arrow) and cell–cell contacts (yellow arrow) are indicated. Higher magnification (center) show contacts (green arrows) interspersed between gaps (red arrows). Base of stitch contacts appear raised (white arrows), indicating tethers are taut (right).

*Figure 5 continued on next page*

*Figure 5 continued*

(**C**) A cell-cell contact (green arrow) compatible with cadherin-based adhesion (~30 nm). (**D**) Interstitial gaps in explants. Labelled (mRFP) explants (left) in medium with AvidinFITC to visualize gaps (red arrows). Stitch contacts (yellow arrows) extend between cells (center). Magnified view of contacts (right). (**E**) Quantification of intercellular distance in explants, whole embryo SEM, and TEM. Measurements were taken from the central, mid-endodermal region. (**F**) Interstitial gaps contain extracellular matrix. Putative heteroglycans stained using Alcian Blue appear as black cell surface residues under TEM (blue arrows) or link cells (yellow arrow). Plot shows data cumulatively sampled from four embryos collected from different egg batches. (**G**) Cells form dynamic intercellular contacts. Membrane label (mGFP; top) and Lifeact-Ruby (middle) co-expressing cells show contacts containing F-actin (merged; bottom). Time-lapse sequence (four right panels) of a region of interest (box) shows that protrusions extend (yellow arrows) and retract (green arrows). Region of interest (red box) is indicated in the top right corner of select panels.

DOI: https://doi.org/10.7554/eLife.27190.016

The following source data is available for figure 5:

**Source data 1.** Quantification of intercellular distance.

DOI: https://doi.org/10.7554/eLife.27190.017

*1*). Our results show that endoderm cell interaction with FN is sufficient for translocation *in vitro*, and necessary for cell movement within the tissue.

## C-cadherin is required for endoderm cell migration

The sites of close (20–30 nm) cell–cell contacts in the endoderm are compatible with cadherin binding. C-cadherin is the main isoform in the early gastrula (*Kühl and Wedlich, 1996*). Its knockdown by a well characterised, C-cadherin mRNA rescuable morpholino antisense oligonucleotide (CcadMO; *Ninomiya et al., 2012*) caused rounding of cells in vegetal slice explants (*Figure 7A–D*; *Figure 7—source data 1*). This shape change was unexpected given that endoderm cell–cell contact is largely mediated through short-lived, lateral filiform protrusions. However, the behaviour of α-catenin indicated that C-cadherin was indeed functional in these transient contacts. In the *X. laevis* gastrula, α-catenin is associated with the cadherin–β-catenin complex at sites of adhesion (*Kurth et al., 1999*). We found that α-catenin was indeed recruited upon contact, but not only to the tips of filiform protrusions. It also accumulated at their bases, outside of the direct contact area, and it dispersed after protrusion retraction (*Figure 7A–B*; *Figure 7—figure supplement 1*). This response suggests that cadherins of adjacent cells transiently engaged in binding.

Strikingly, CcadMO-injected cells possessed fewer lateral protrusions (*Figure 7A–B,E*; *Figure 7—source data 1*). Those that formed were as stable as in controls, but protrusion initiation itself was reduced upon C-cadherin knockdown (*Figure 7F,G*; *Figure 7—source data 1*). In turn, less α-catenin accumulated laterally in cells as fewer protrusions were present at any point in time in morphant cells (*Figure 7A–B*). A feedback between cadherin-dependent α-catenin recruitment and filiform protrusion formation appeared to be involved in endoderm cell interaction. Ultimately, this process amounted to strengthening of cell–cell adhesion. CcadMO-injection reduced tissue surface tension, a measure of tissue cohesion (*Winklbauer, 2015*) in the endoderm (*Figure 7H*; *Figure 7—source data 1*). Thus, although contacts compatible with cadherin adhesion were infrequent and minute, C-cadherin contributed significantly to the mutual attachment of vegetal endoderm cells.

C-cadherin-based interaction is essential for endoderm cell migration. In vegetal slice explants, cell trajectories were reduced by CcadMO to the level of the most vegetal cells in normal explants (*Figure 7I–K*; *Figure 7—source data 1*). Paradoxically, it was exactly in the regions where packing was less dense that C-cadherin was required for cells to elongate and to attain their full migration velocity. Our results indicated that in these regions, interactions of cells with the ECM component FN and through the cell adhesion receptor C-cadherin were both required for amoeboid differential migration.

## Trailing edge retraction involves ephrinB1-dependent macropinocytosis and *trans*-endocytosis

We noted a unique mechanism of tail retraction in endoderm cells. The process appeared as a simple narrowing of the cell–cell contact area at the rear end (*Figure 3A,D*) when viewed in a plane outside the actual retracting rim (*Figure 8A,B*). If the complete tail was exposed, however, it resembled a lamelliform protrusion (*Figure 8A,B*) that was actin-rich and actively protruded and retracted while attached to the surface of an adjacent cell (*Figure 8C,D*; *Videos 5* and *6*). However, the protrusion

**Figure 6.** Fibronectin is required for vegetal endoderm cell migration. (**A**) Endoderm cell morphology on different substrates. Cells on plastic coated with FN (left) are multipolar (red arrows). Cells on gelatin coated with FN (right) are unipolar with front (yellow arrow) and rear (blue arrow). (**B**) Morphology *in vivo*. Endoderm cell with front (yellow arrow) and rear (blue arrow) polarity. Animal (An) is up, vegetal (Vg) is down. (**C**) Cell spreading on FN. Cell shape changes (top row) are outlined (bottom row), consecutive shapes (grey) differ from shapes at previous time points (difference in black).
*Figure 6 continued on next page*

*Figure 6 continued*

Time in minutes is indicated. (**D**) Cell locomotion on gelatin-FN *in vitro*. Morphological changes (top row) are outlined (mid row) along with movement direction (black arrows) and differences in cell shape (black) between time points. A representative cell is followed, panels show bursts of movement over the course of 45 min. Interpretation of cell behaviours (bottom row). (**E**) Cell migration velocity *in vitro*. Velocities of cells in explants with respect to the epithelium, and of single cells with respect to *in vitro* substrates. (**F**) Inhibition of FN binding in embryos. Embryos were injected with RGD (left) or RGE peptides (center) into the blastocoel at blastula stage 8, or left uninjected (right), cultured until stage 11, fixed, and sectioned sagittally. RGD treatment perturbed endoderm morphology relative to controls (yellow arrows). (**G**) Inhibition of FN binding in explants. Explants in medium containing RGD (left), or RGE (right) peptides. RGD-treated cells appear rotund (pink outline), RGE-treated cells elongated (yellow outline). Region of interest (red box) is indicated in the top right corner of select panels. (**H**) Cell length-width ratio in explants incubated in RGE (left) or RGD peptides (right). (**I**) Cell elongation congruity (defined in *Figure 6—figure supplement 1*) in explants incubated in RGE (left) or RGD peptides (right). (**J**) Cell migration velocity in explants incubated in RGE (left) or RGD peptides (right). For H–J, plots show data sampled from three embryos from different egg batches.
DOI: https://doi.org/10.7554/eLife.27190.018

The following source data and figure supplement are available for figure 6:

**Source data 1.** Quantification of cell migration velocity and congruity.
DOI: https://doi.org/10.7554/eLife.27190.020
**Figure supplement 1.** Schematic of morphometric analyses.
DOI: https://doi.org/10.7554/eLife.27190.019

pointed opposite to the direction of cell movement, and was dragged behind the advancing cell body while accumulating localised membrane clusters (*Figure 8D*; *Video 5*). Increased membrane undulations were also seen more laterally at the trailing edge, and FITC-conjugated dextran was taken up in vesicles that initiated from membrane pits (*Figure 8E*). When viewed under TEM, vesicles were found in cell tails in densely packed clusters (*Figure 8F*), and structures consistent with different stages of endocytosis were discernable at the membrane (*Figure 8G*). Trailing edge vesicles were also seen in the SEM where the cell surface was incidentally broken, whereas the intact surface showed pits of a similar size (*Figure 8—figure supplement 1*).

Vesicle sizes varied from 0.05 to 3 μm (*Figure 8H*; *Figure 8—source data 1*), which well exceeded the 0.1–0.2 μm range for clathrin-mediated endocytosis (*McMahon and Boucrot, 2011*), but were consistent with formation by macropinocytosis. Rab5 is associated with early stages of macropinocytosis (*Lanzetti et al., 2004*), and it was enriched in the trailing domain. Rab5-CFP puncta accumulated and resolved on the scale of minutes alongside membrane clusters that entered the cytoplasm (*Figure 8I*; *Video 7*). The association of vesicle internalisation with high protrusive activity, the large and heterogeneous size of vesicles, and their interaction with Rab5 strongly suggest that endocytosis at the trailing edge was based on macropinocytosis. Thus, translocation of the endoderm cell rear occurred by the forward movement of the yolk-rich content of the cell body, which left behind a lamelliform, highly protrusive tail that was, at least partially, resorbed by macropinocytosis.

EphrinB, a transmembrane protein signalling ligand, has been implicated in macropinocytosis (*Bochenek et al., 2010*), and *X. laevis* endoderm expresses a full complement of EphB and ephrinB isoforms (*Rohani et al., 2011*). Because we had noted an effect of ephrinB1 on vegetal rotation (*Figure 9—figure supplement 1*), we examined its localisation in migrating cells using a fluorophore-fused construct, ephrinB1-mCherry. Immediately after explantation, ephrinB1-mCherry was evenly expressed at the cell membrane, but as cells attained an elongated morphology, it progressively accumulated at the rear membrane (*Figure 9A*; *Video 8*). Enrichment at the trailing edge was maintained during migration (*Figure 9B,C*; *Figure 9—figure supplement 2*; *Figure 9—source data 1*). In ephrinB1-MO morphant cells, endosome number in the trailing domain was decreased (*Figure 9E*; *Figure 9—source data 1*). Overexpression of ephrinB1 increased endosome numbers (*Figure 9E*) and induced ectopic vesicle internalisation

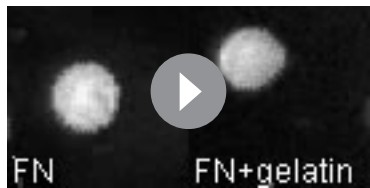

**Video 4.** Endoderm cell migration *in vitro*. Cell behaviours on FN-adsorbed tissue culture plastic (left), and FN-adsorbed gelatin coated dish (right). Movie shows autonomous cell behaviours over the course of 10 min.
DOI: https://doi.org/10.7554/eLife.27190.021

**Figure 7.** C-cadherin is required for endoderm cell migration. (**A**) Protrusions engage in cell-cell adhesion. Labelled (mRFP) endoderm cells are elongated (yellow outline) in explants (left). Cells project protrusions (yellow arrows) onto neighbouring cells (center). Protrusions are enriched with α-catenin (blue arrow) at sites of cell-cell contact (right). (**B**) C-cadherin knockdown altered cell morphology and reduced protrusion formation. C-cadherin morpholino (CcadMO) injected cells appear rotund (pink outline) (left) and extend few protrusions (center). However, α-catenin (blue arrow) was still

*Figure 7 continued on next page*

*Figure 7 continued*

present within protrusions (right). Region of interest (red box) is indicated in the top right corners. (C) Cell length-width ratio in uninjected and CcadMO-injected explants. Cells were sampled from the middle of the central column of the endoderm. (D) Cell elongation congruity in uninjected and CcadMO-injected explants. For C and D, plots show data sampled from three embryos from different egg batches. (E) Average number of protrusions per cell in uninjected and CcadMO-injected explants. For all histograms, error bars indicate S.E. (F, G). Protrusion dynamics of uninjected and CcadMO-injected cells. Zero indicates no change (dash grey line), net gain (positive axis) and net loss (negative axis) of protrusions between consecutive time points are shown. Colours represent individual cells. For E–G, plots show data from three embryos from different egg batches. (H) Quantification of tissue cohesion. Tissue surface tension measured from uninjected and CcadMO-injected endoderm. Plot shows data from 22 embryos from different egg batches. (I) Cell migration velocity in explants comparing uninjected and CcadMO-injected cells. (J, K) Movement trajectories of cells in explants. (J) Uninjected, (K) CcadMO-injected explant after 60 min. Colours represent individual cells. For I–J, plots show data from four embryos collected from different egg batches.

DOI: https://doi.org/10.7554/eLife.27190.022

The following source data and figure supplement are available for figure 7:

**Source data 1.** Quantification of the effects of C-cadherin knockdown on cell morphology and behaviour.

DOI: https://doi.org/10.7554/eLife.27190.024

**Figure supplement 1.** Localization of α-catenin.

DOI: https://doi.org/10.7554/eLife.27190.023

throughout the cell surface, leading to cell rounding and detachment (*Figure 9D,H*), although cells still showed a 'kneading' motion of their surface. In summary, we concluded that ephrinB1 regulated endocytosis in *X. laevis* endoderm cells.

Despite being largely separated from cells laterally (*Figures 5B* and *8F*), the elongated shape of cell tails indicated attachment at some point. The detachment required for eventual tail retraction could also be mediated by ephrinB1, and a double role was in fact indicated by the presence of two types of vesicles. At interstitial spaces, large vesicles pinching off from the cell surface co-localised with ephrinB1-mCherry (*Figure 10A*), and consistent with the cytoplasmic attachment of the mCherry tag, vesicles were labelled on their cytoplasmic surfaces (*Figure 10B*). However, some vesicles showed more intense membrane labelling and had the mCherry label on both the inner and outer surfaces (*Figure 10C*). This finding suggested that the vesicles were generated by *trans*-endocytosis, a mechanism whereby an ephrin/Eph receptor-associated cell contact is resolved; a cell endocytoses its own membrane together with that of the adjacent cell to which it is linked through the ephrin/Eph interaction (*Gaitanos et al., 2016*). Indeed, we observed in the TEM, in addition to numerous simple vesicles, a smaller number of double-layered ones, that is, cytoplasm-filled inner vesicles within outer vesicles (*Figure 10E*), as expected from *trans*-endocytosis (*Figure 10C*).

Whereas macropinocytosis removes free cell surfaces, *trans*-endocytosis can break cell contacts at the rear, and remove surfaces in contact with adjacent cells. Together, ephrinB1-dependent macropinocytosis and *trans*-endocytosis could permit tail resorption and retraction in endoderm cells (*Figure 10D*). In fact, endocytosis at the cell rear was correlated with trailing edge retraction. Normally, the width of the trailing edge decreased continually, whereas in ephrinB1-MO morphant cells, it remained unchanged (*Figure 9F,G*; *Figure 9—source data 1*). Consistent with impaired tail retraction, ephrinB1-MO morphant cells became elongated (*Figure 9H*), and vegetal rotation was halted (*Figure 9—figure supplement 1*). We propose that resorption of the tail by endocytosis is an essential component in the retraction of the trailing edge of vegetal endoderm cells, which in turn is necessary for their migration. However, we cannot exclude the possibility that ephrinB1 also functions outside the trailing edge to modulate cell contact behaviours required for translocation.

## Discussion

Endoderm internalisation in *X. laevis* has been ascribed to a unique gastrulation movement, vegetal rotation. To determine how it is related to other, more common gastrulation mechanisms, we analysed its cellular basis. Our results depict vegetal rotation as an adaptation of a more ancient, epithelial-type morphogenetic process to the multilayered structure of gastrula tissues in vertebrates, and place it in the context of endoderm internalisation in other metazoans. We argue that endoderm cell movement strikingly resembles the ingression of cells during gastrulation in other organisms, except



**Figure 8.** Trailing edge retraction by rear-end membrane remodeling and macropinocytosis. (**A**) Regions of interest. Schematic illustrates endoderm cells in a vegetal explant, viewed through a glass slide with non-adhesive BSA coating. Direction of migration is to the left. Region 1 (ROI1) shows the plane of focus at the level of the surface of the outermost cells, and region 2 (ROI2) a plane through the cell bodies. ROI1 and ROI2 are approximately 0.5 μm apart. (**B**) Trailing edge representation at different planes of focus. The cell rear of a membrane labelled endoderm cell within the vegetal

*Figure 8 continued on next page*

*Figure 8 continued*

explain is shown. Images represent ROI1 and ROI2 introduced in (A). Membrane protrusions (arrows) are clearly visible at the substrate level. (C) Tail retraction of cell co-expressing mRFP and Lifeact-GFP. (D) Cluster of vesicles (yellow arrow) is visible at the trailing edge (top row). Enlargement of inset (box) region for all panels of the sequence (bottom row). (E) Uptake of extracellular fluid at the trailing edge (top row). Membrane label (mRFP) and probe (dextran) are shown separately in black and white (middle panels). Vesicle of interest (yellow arrow) is indicated. A nearby yolk platelet (Y) is noted. Sequence of process is illustrated below. (F) Ultrastructure of endoderm cell tail in the embryo. Overview (left), higher magnification (i) of the trailing edge shows vesicle clusters (yellow arrows). (G) Ultrastructure of vesicles near the rear membrane (left). Putative stages of vesicle internalization (right, top row) and corresponding illustrated interpretations (red arrows; bottom row) are shown. (H) Quantification of vesicle sizes from the trailing edge at different gastrula stages. Box plots show the median, interquartile range, maximum and minimum. Data were cumulatively sampled from 12 embryos collected from different egg batches. (I) Rab5c-CFP is enriched at the trailing edge during migration (left panels). Magnification of the cell-rear (inset i–iii) shows that Rab5 is localized to sites of prominent membrane remodeling (yellow arrows).

DOI: https://doi.org/10.7554/eLife.27190.025

The following source data and figure supplement are available for figure 8:

**Source data 1.** Quantification of vesicle diameter.
DOI: https://doi.org/10.7554/eLife.27190.027
**Figure supplement 1.** Scanning electron micrograph of cell trailing edge *in vivo* (left).
DOI: https://doi.org/10.7554/eLife.27190.026

that it occurs in the deep cells of a multilayered tissue. We have termed the underlying cell behaviour ingression-type migration.

## Endoderm internalisation by ingression-type, differential cell migration

In invertebrates, internalisation of mesoderm and endoderm starts from a single-layered cell array, typically an epithelium. In the sister group to vertebrates, the urochordates, a single layer of endoderm cells invaginate by constricting apically and laterally, thus inverting its wedge shape (*Figure 11A*) (*Satoh, 1978*; *Sherrard et al., 2010*). In *X. laevis*, the corresponding region has increased dramatically in size. The large vegetal blastomeres also initially form a single-layered array, but soon become subdivided into a multilayered vegetal cell mass by periclinal and anticlinal cleavage divisions (*Figure 11B*). Nevertheless, the vegetal endoderm as a whole performs a similar inversion of its wedge-shaped configuration to narrow at the external surface and to widen inside the embryo (*Bauer et al., 1994*) (*Figure 11B*). This tissue deformation is based on deep cell rearrangement. The outer, epithelial layer does not invaginate except in a small zone at its margin where bottle cells form (*Figure 11B*). Otherwise, the epithelium constricts moderately to accommodate the narrowing of the vegetal mass (*Keller, 1978*), and later bends upward at its periphery during archenteron formation (*Evren et al., 2014*) (*Figure 11B*). However, the epithelial layer retains its integrity, and cells do not leave the surface by ingression (*Keller, 1978*).

Deep vegetal endoderm cell rearrangement is due to the migration of cells over each other without the use of lamellipodial, filopodial or bleb-like protrusions; thus, we have classified this mode of migration as amoeboid (*Figure 11E*). Amoeboid migration is typically seen when single cells move through three-dimensional ECM or between stationary cells, as documented for human leukocytes, zebrafish primordial germ cells, and cancer cells (*Mandeville et al., 1997*; *Sahai and Marshall, 2003*; *Wolf et al., 2003*; *Blaser et al., 2006*). *X. laevis* vegetal endoderm provides an example of amoeboid translocation in the context of collective cell movement (*Figure 11D*). Previously, *Holtfreter (1944)* noted that in the salamander *Ambystoma*, prospective endoderm cells take on a 'sausage shape' when isolated, and engage in an 'obscure gliding movement'. Amoeboid creeping of elongated endoderm cells on agarose was also observed for the newt *Cynops pyrrhogaster* (*Kubota, 1981*), and 'sausage-like' cells were isolated from the endoderm of the frog *Rana pipiens* (*LeBlanc et al., 1981*). These findings suggest that amoeboid migration is a widespread mechanism for amphibian endoderm internalisation.

Several sub-types of amoeboid migration can be distinguished (*Paluch et al., 2016*), but the specific mechanism of amphibian endoderm cell translocation is not known. Amoeboid translocation is typically associated with low substratum adhesion and movement in confined spaces where traction is generated by repeated cell shape changes (*Paluch et al., 2016*; *Te Boekhorst et al., 2016*). Vegetal endoderm is indeed the least cohesive tissue in the gastrula (*David et al., 2014*), and cells cycle through elongation of the cell body, bulging of the cell front, and narrowing and retraction of the

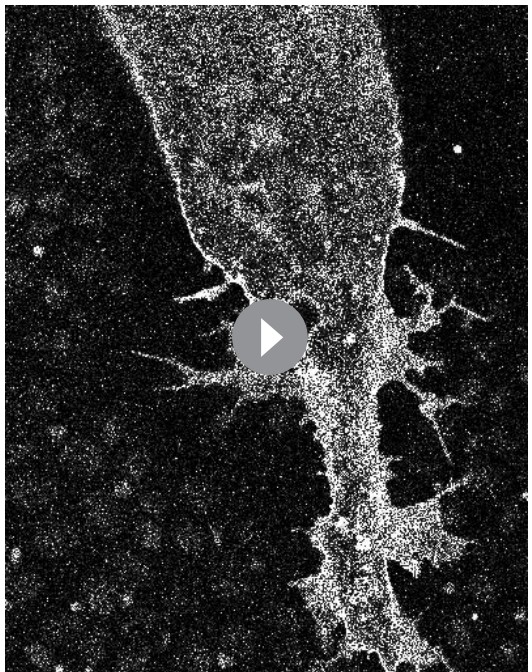

**Video 6.** Trailing edge undulation is associated with membrane protrusion formation and resolution. High-resolution time-lapse video shows numerous lamelliform projections extending outward from the cell body and then retracting. Membrane-bound vesicles could be observed at sites of membrane fluctuation which are taken up into the cytoplasm.
DOI: https://doi.org/10.7554/eLife.27190.029

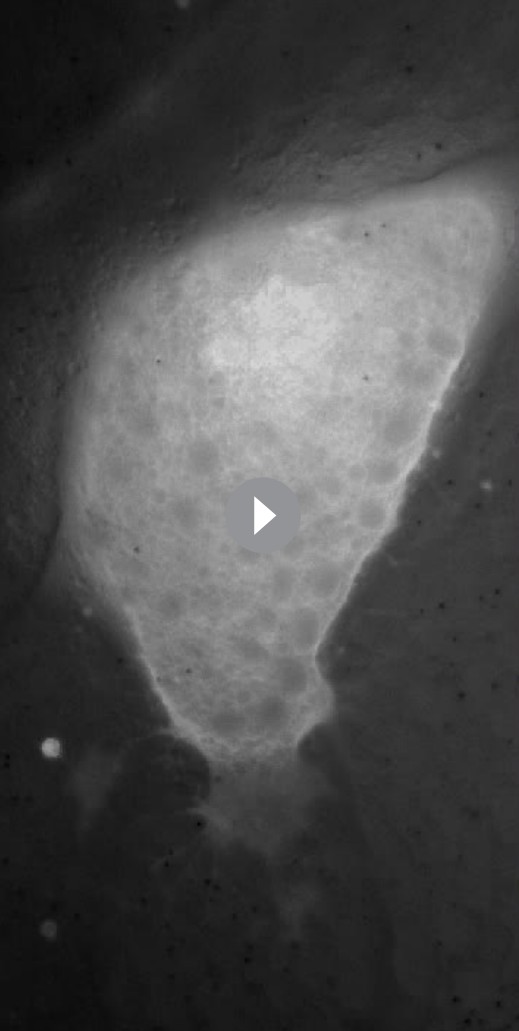

**Video 5.** Cell rear retraction during translocation is associated with trailing edge membrane undulation. Time-lapse video shows a membrane-labelled endoderm cell moving animally between unlabelled endoderm cells within a mosaic-labelled vegetal explant.
DOI: https://doi.org/10.7554/eLife.27190.028

trailing edge (*Figure 11E*). *Ambystoma* endoderm cells also employ shape changes *in vitro* (*Holtfreter, 1944*), whereas *C. pyrrhogaster* cells glide continuously (*Kubota, 1981*). However, all of these cells can move in isolation while lightly attached to a surface, without the need for lateral confinement, which excludes mechanisms such as 'chimneying' (*Yip et al., 2015*). Moreover, cells move their yolk-filled contents forward relative to sites of substratum attachment, but no uropod pushes the cytoplasm by myosin II-powered contractions, and the yolk platelets show no signs of flowing within cells in a fountain movement; instead, they rigidly maintain their relative positions. In *C. pyrrhogaster* endoderm cells, membrane flows continuously from the front to the rear end, relative to a yolk-filled interior that behaves as a coherent body (*Kubota, 1981*). Further study will be necessary to identify the mechanism that moves the cell content forward relative to the substratum-attached membrane.

The most striking difference from other instances of amoeboid migration is seen at the cell's rear end. Instead of being pulled forward by a uropod (*Vicente-Manzanares et al., 2007*; *Lämmermann et al., 2008*), the trailing edge of *X. laevis* endoderm cells is at least in part resorbed by massive macropinocytosis. Likewise, in *C. pyrrhogaster* endoderm cells, large vesicles accumulate and membrane protrusions are present at the rear end (*Kubota, 1981*). The combination of locomotion by amoeboid leading-edge behaviour and macropinocytotic tail retraction links endoderm locomotion more closely to bottle cell invagination and ingression than to other examples of amoeboid migration (*Figure 11C,D*).

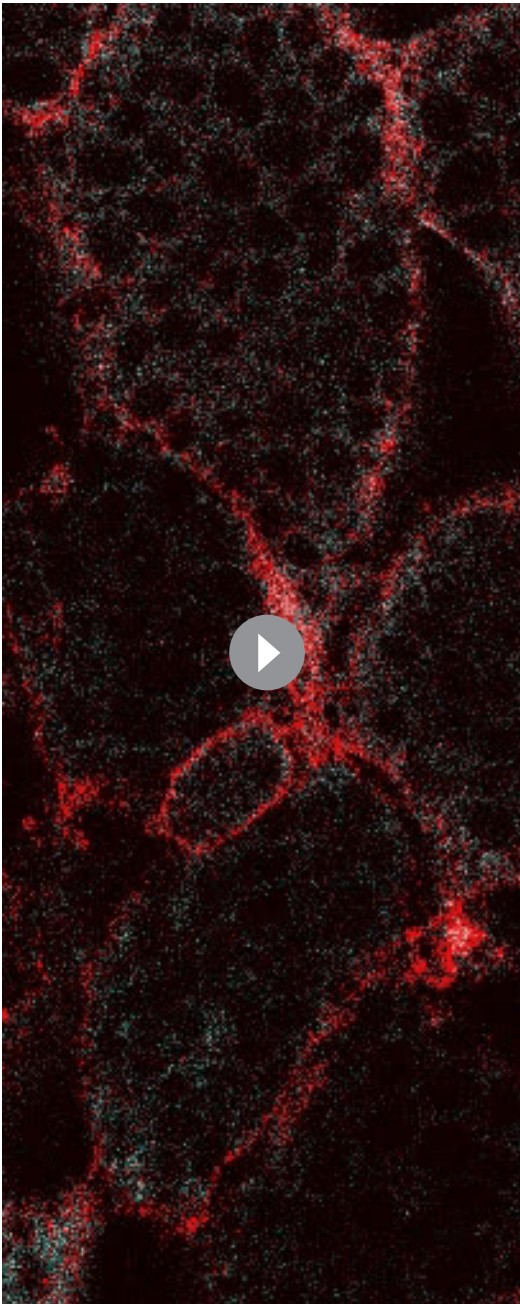

**Video 7.** Cells co-expressing Rab5c-CFP and membrane-RFP in explant. An animally-located endoderm cell is seen detaching from a vegetally-located cell. Rab5 enrichment could be observed at the trailing edge of the retracting cell.
DOI: https://doi.org/10.7554/eLife.27190.030

Invagination by bottle cell formation, and gastrula-stage ingression in the sea urchin or at the primitive streak of birds and mammals, start similarly. The basal end of cells bulges interiorly while the apical portion narrows (*Figure 11C*). This is the end state in bottle cell invagination. Ingressing cells continue to move to the inside, across like cells from which they are separated by ECM-filled gaps, without using locomotory protrusions, until they detach from the surface, retract their necks, and leave the epithelial layer (*Figure 11C*) (*Balinsky and Walther, 1961*; *Perry et al., 1966*; *Granholm and Baker, 1970*; *Wakely and England, 1977*; *Batten and Haar, 1979*; *Katow and Solursh, 1980*; *Komazaki, 1995*; *Viebahn et al., 1995*; *Lee and Harland, 2010*; *Williams et al., 2012*). During this process, the apical cell surface forms actin-rich protrusions and internalises large vesicles, which often become packed into clusters. This process resembles macropinocytosis, which can occur at the apical membrane of epithelial cells (*Mettlen et al., 2006*). In addition, cytoplasm-filled vesicles-within-vesicles occur in bottle cells of *X. laevis* and the newt *Triturus* (*Perry et al., 1966*; *Lee and Harland, 2010*), and the pinching-off of such a vesicle was captured in a mouse primitive streak cell (*Batten and Haar, 1979*). Based on their co-occurrence with labelled ephrinB1-containing vesicles, we tentatively identified similar structures in *X. laevis* endoderm as the products of *trans*-endocytosis. This mechanism drives cell–cell separation upon Eph receptor–ephrin interaction – these molecules actually form strong bonds – by one cell engulfing the interaction site including the membrane of the adjacent cell (*Gaitanos et al., 2016*).

The high protrusive activity associated with intense vesicle formation at the trailing edge of endoderm cells, combined with the amoeboid translocation of the cell body, suggests that vegetal cell migration may be derived from an invagination or ingression process (*Figure 11C,D*). A further commonality between endoderm cell migration and ingression is that cells use each other as substratum for translocation in a collective cell movement. Whereas amoeboid migration in general is considered to be independent of specific cell adhesion mechanisms, *X. laevis* endoderm cells require interaction with FN and cadherin for proper migration.

Bottle cells form transiently or permanently during ingression or invagination, respectively, and both processes are mechanistically related. For example, whereas mesoderm invaginates in *D. melanogaster*, it ingresses in the midge *Chironomus riparius*, and a change in the expression of one or two effector genes of Snail is sufficient to switch between the two mechanisms (*Urbansky et al., 2016*). In vegetal endoderm cells, the stereotypical motile behaviour characteristic of ingression is executed repeatedly, constituting a special case of

**Figure 9.** EphrinB1 involvement in endosome biogenesis. (**A**) Time-lapse sequence of ephrinB1-mCherry distribution following explantation. After 5 min, ephrinB1 (eB1) has become enriched at the trailing membrane (white arrows). (**B**) Co-localization of ephrinB1 with endosomes (top row). Higher magnification (bottom row, i–iii) shows that ephrinB1 is enriched near endosomes (white arrow). (**C**) Quantification of ephrinB1-mCherry fluorescence intensity relative to mGFP at different plasma membrane domains. Bars indicate S.E. (**D**) Overexpression of full-length ephrinB1 (eB1FL) in membrane-

*Figure 9 continued on next page*

*Figure 9 continued*

labelled cells causes aberrant vesicle formation at the entire cell cortex (white arrows). (E) Quantification of vesicle number at the trailing edge of uninjected (control), ephrinB1-morpholino (eB1MO) injected, and eB1FL mRNA-injected cells. (F) Trailing edge membrane tapering during retraction. Uninjected cells show typical recession behaviour (top row), while the rear of eB1MO-injected cells remains blunt (bottom row). (G) Quantification of trailing edge width in uninjected (left) and eB1MO-injected (right) cells. Average rate change is shown (black line). Colours represent individual cells. (H) Morphology of uninjected (left), eB1MO-injected (center), and eB1FL mRNA-injected (right) labelled (mRFP) cells in live explants. Animal is to the top, vegetal to the bottom. For C, E, and G, plots show data sampled from five embryos collected from different egg batches.

DOI: https://doi.org/10.7554/eLife.27190.031

The following source data and figure supplements are available for figure 9:

**Source data 1.** Quantification of ephrinB1 distribution, effects of ephrinB1 on vesicle number and trailing edge width.

DOI: https://doi.org/10.7554/eLife.27190.034

**Figure supplement 1.** Mid-sagittal fractures of stage 12 uninjected (left), eB1MO-injected (centre), and eB1FL mRNA-injected (right) gastrulae.

DOI: https://doi.org/10.7554/eLife.27190.032

**Figure supplement 2.** Co-localization of ephrinB1 (eB1-mCh) with membrane-label (mGFP) shows that ephrinB1 is dispersed over the entire membrane, but enriched at the trailing edge membrane (yellow arrows), particularly at the rear (blue arrows).

DOI: https://doi.org/10.7554/eLife.27190.033

amoeboid translocation, ingression-type migration. Ingression proper is absent from the vegetal epithelial layer (*Keller, 1978*), and we propose that vegetal rotation is derived from invagination. Apical constriction of vegetal epithelial layer cells was occasionally observed in the SEM. Because deep vegetal cells do not possess an apical domain through which they would be linked, they simply move past each other when executing their program of bottle cell formation (*Figure 11C,D*).

Endoderm migration is directional, with cells initially polarised in the A–V direction, which suggests that during the cleavage divisions that generate the multilayered structure of the vegetal mass, the primary apical–basal polarity of the vegetal blastomeres is transmitted to all deep cells by an unknown mechanism. Animal–vegetal polarisation of the vegetal mass is apparent in the graded cell packing density along this axis. Moreover, numerous genes show A–V differences in expression at the initial gastrula stage (*Taverner et al., 2005*). For example, a hyaluronic acid synthase is expressed in the animal half of the vegetal cell mass, consistent with hyaluronan supporting the extended interstitial space in this region. Likewise, the Wnt/planar cell polarity components Frizzled-7 and Prickle are enriched in the animal part of the region. Whether cells are also individually polarised or gain polarity and orientation cues during migration from these spatial inhomogeneities remains unclear. Endoderm cells migrating on FN-gelatin are polarised, but we do not know whether this polarity is identical to that within the tissue, or generated by spontaneous symmetry breaking *in vitro*. The cues that re-orient endoderm cells as they approach the BCF likewise remain to be identified.

Cell rearrangement by junction remodelling in epithelial monolayers has been extensively studied, but less detailed information is available for the corresponding processes in multilayered tissues. Differential migration may be an archetypical mode of rearrangement in such compact tissues. Mesoderm involution (*Evren et al., 2014*) and the radial intercalation of prechordal mesoderm (*Damm and Winklbauer, 2011*) are examples of this mechanism in the *X. laevis* embryo. Likewise, the dorsal migration of lateral mesoderm in zebrafish involves differential migration (*Roszko et al., 2009*; *Yin et al., 2009*), and fin or limb bud morphogenesis in fish and mice, respectively, depend on three-dimensional patterns of cell migration (*Ede et al., 1974*; *Wyngaarden et al., 2010*; *Mao et al., 2015*). Vegetal rotation is another example of cell rearrangement by differential migration. Appropriately graded velocity differences have been observed in explants, and time-lapse X-ray microtomography revealed similar velocity gradients in the embryo (*Moosmann et al., 2013*).

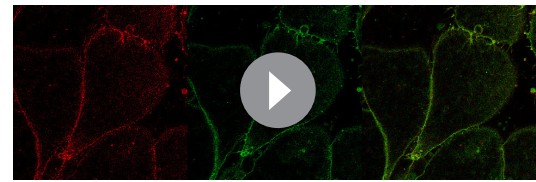

**Video 8.** EphrinB1 is enriched at the trailing edge membrane and is present within the membrane of internalized vesicles at the cell rear. For clarity, panels show ephrinB1-mCherry alone (left), membrane-GFP alone (center), and the corresponding merged composite (right).

DOI: https://doi.org/10.7554/eLife.27190.035

**Figure 10.** Macropinocytosis, *trans*-endocytosis and cell detachment. (**A**) Macropinosome formation. Sequence shows membrane undulations of an eB1-mCh and mGFP co-expressing cell at the trailing edge. A membrane indentation is pinched off to form an endosome (arrows). Separate panels are shown for mGFP and eB1-mCh, an interpretation of the process at the bottom. (**B**) Single membrane vesicle. A vesicle in the trailing edge cytoplasm of an eB1-mCh and mGFP co-expressing cell shows eB1-mCh at the outside. An interpretative illustration is shown (right). (**C**) Double membrane vesicle. A vesicle located in the trailing edge cytoplasm of an eB1-mCh and mGFP co-expressing cell shows label on both sides of the vesicle membrane. An interpretation is shown (right). (**D,E**) Membrane endocytosis during cell-cell contact resolution. (**D**) Sequence shows cell detachment of mGFP and ephrinB1-mCh co-expressing cells. A leader cell (upper) resolves its contact (arrows) with a follower (lower) cell. Composite and individual channels are shown. Numerous vesicles are seen at the site of contact during detachment in the leading cell. An interpretation of the process is shown (bottom). (**E**) TEM of putative *trans*-endocytotic vesicles. (i) A leading cell presumably undergoing membrane *trans*-endocytosis as it detaches from a follower cell. A

*Figure 10 continued on next page*

*Figure 10 continued*

double layered vesicle containing cytoplasmic cargo is indicated (arrow). (ii) Another example of a cytoplasm-filled vesicle within a vesicle (arrow) in the trailing edge cytoplasm. (iii) Various vesicles (arrows) within the trailing edge cytoplasm.

DOI: https://doi.org/10.7554/eLife.27190.036

### Cell–cell interaction in the vegetal cell mass

Like ingressing cells in the sea urchin, chicken or mouse (*Balinsky and Walther, 1961*; *Granholm and Baker, 1970*; *Wakely and England, 1977*; *Batten and Haar, 1979*; *Katow and Solursh, 1980*; *Komazaki, 1995*; *Viebahn et al., 1995*; *Lee and Harland, 2010*; *Williams et al., 2012*), translocating endoderm cells in *X. laevis* and other amphibians (e.g. *Nakatsuj and Nakatsuj, 1975*) are separated by prominent interstitial gaps. ECM material is present in the gaps, but a self-supporting, *trans*-cellular ECM structure is unlikely to permeate the endoderm for several reasons. First, when fixed for TEM, the ECM collapses locally into densely stained, isolated globules at the surface of cells, and when endoderm is dissociated in calcium-free medium, no conspicuous, stable ECM scaffold remains. Secondly, vegetal explants deform rapidly beyond their initial contours by cell migration, a behaviour that is difficult to reconcile with a stable, pre-formed matrix scaffold as an

**Figure 11.** Ingression-type cell migration during *X. laevis* vegetal rotation. (A) Endodermal cells (yellow) invaginate in ascidians. Schematic of *Ciona intestinalis* embryos at 64 and 112 cell stages. (B) Vegetal rotation in amphibians. Schematic of *Xenopus laevis* embryos at 32 cell, late blastula, and mid-gastrula stages. Endoderm cells of the vegetal endoderm, the suprablastoporal endoderm, and bottle cells are shown in yellow. Inset indicates cells shown in (D). Archenteron (arc), dorsal bottle cells (dbc), and ventral bottle cells (vbc) are indicated. (C) Generalized schematic of epithelial cell ingression shows ingressing cells (yellow) next to non-ingressing cells (grey). Internalized membrane vesicles are shown at the trailing edges. (D) Ingression-type cell migration. Schematic shows endoderm cells undergoing differential amoeboid migration in the vegetal cell mass. (E) Endoderm cells rearrange by cycling through a series of amoeboid migration behaviours (indicated by dashed lines) which include cell body elongation, cell front expansion in tandem with cell rear narrowing which is required for trailing edge retraction.

DOI: https://doi.org/10.7554/eLife.27190.037

external substratum. In addition, the movement of individual endoderm cells is the resultant of active and passive components, which is inconsistent with individual cells migrating through a stable, external ECM scaffold. Lastly, proteins that form covalently cross-linked ECM structures, such as collagens, laminin or fibrillin, are not expressed in the early gastrula (*Winklbauer and Ettensohn, 2004*).

Thus, the ECM in the vegetal cell mass is best viewed as a cell surface coat that contributes to weak and flexible adhesion between cells. A highly hydrated, space-filling constituent of the ECM in the *X. laevis* gastrula is hyaluronic acid (*Müllegger and Lepperdinger, 2002*), which may act as a spacer between cells, whereas FN contributes to cell adhesion. FN and its receptor α5β1 integrin are expressed in gastrula endoderm (*Winklbauer and Ettensohn, 2004*), and the knockdown of FN diminishes cell–cell contacts in the endoderm (*Barua et al., 2017*). Likewise, inhibition of FN–integrin interaction with RGD peptide leads to vegetal cell detachment. FN-mediated cell–cell adhesion has been demonstrated for other cell types (*Robinson et al., 2003*, *2004*). FN inhibition impedes endoderm cell migration and affects internalisation of the vegetal endoderm (*Ramos and DeSimone, 1996*; *Winklbauer and Keller, 1996*), which may be a consequence of diminished adhesion, or due to reduced integrin signalling.

Vegetal endoderm cells are also connected via small direct membrane contacts and filiform lateral protrusions. Despite the infrequent occurrence of cell–cell contact, C-cadherin is required for proper endoderm cell migration. Cadherins couple cells mechanically, but also act as signalling molecules that control the actin cytoskeleton (*Priya and Yap, 2015*). During cell rearrangement by junction remodelling, both roles are important to drive neighbour exchanges in cells closely attached to each other through adherens junctions. In the loosely packed endoderm, knockdown of C-cadherin impedes lateral cell–cell contacts through attenuation of filopodia formation. Cells round up and fail to translocate, but tissue cohesion is still retained; we speculate that here C-cadherin mainly assumes a signalling role. Cadherin-based contact could control the lateral cell cortex through the recruitment of α-catenin (*Drees et al., 2005*; *Amack and Manning, 2012*; *Winklbauer, 2015*), and permit for example the lengthening of the cell body as an essential step in translocation. Altogether, amoeboid migration in the vegetal endoderm is not independent of specific adhesion mechanisms. This difference from other instances of amoeboid translocation may reflect the fact that endoderm cells move not single, but as a coherent mass to rearrange by differential migration.

Interstitial space between migrating amoeboid cells is common in vertebrate embryos. In sections, it often appears as channels between cells that run in parallel to the longitudinal axes of the cell bodies, are of even width, and follow the contours of neighbouring cells, consistent with cell–cell attachment across the gaps (e.g. *Granholm and Baker, 1970*; *Viebahn et al., 1995*). This pattern is also seen in adult tissues of basal metazoans, such as in poriferan archaeocytes (*Weissenfels, 1982*), where cell adhesion is mediated by large, space-filling carbohydrate–protein complexes (*Fernàndez-Busquets and Burger, 2003*). Movement of a space-filling ECM, together with the migrating cells that it surrounds, has been observed, for example, in the chicken gastrula, when internalised mesoderm moves away from the primitive streak (*Vanroelen et al., 1980*; *Van Hoof and Harrisson, 1986*). The amoeboid migration of cells in contact with each other through ECM coats may be an ancestral mechanism of cell rearrangement and morphogenesis in metazoans.

## Materials and methods

### Animal husbandry

Adult *X. laevis* were housed in aquaria, water temperature 19–20°C. Research animals were used in accordance with guidelines approved by the University Animal Care Committee (Protocol no. 20011765, University of Toronto, Canada).

### Embryos

*X. laevis* embryos were obtained by *in vitro* fertilisation (*Winklbauer, 1986*). Briefly, eggs were obtained and fertilised with macerated testes. To remove the jelly coat, embryos were incubated in a solution of 2% cysteine (w/v, pH 8.0) in 0.1X Modified Barth's Saline (MBS; 88 mM NaCl, 1 mM KCl, 2.4 mM NaHCO₃, 0.82 mM MgSO₄, 0.33 mM Ca(NO₃)₂, 0.41 mM CaCl₂, 10 mM Hepes (+NaOH), 1% Streptomycin, 1% Penicillin, pH 7.4) for 5 min and subsequently cultured in 0.1X MBS

until the desired stage for experimentation. Embryos were staged according to *Nieuwkoop and Faber, 1967*.

## Microinjections

Embryos were microinjected in a 3% Ficoll (w/v; Sigma-Aldrich, St. Louis, MO) solution in 1X MBS on plasticine-coated injection dishes. All embryos were seeded into Ficoll dishes 15 min before injection and allowed to heal for 1 hr in Ficoll after injection. All microinjections were performed at the four-cell stage.

## Reagents

### Constructs

Palmitoylated membrane-binding red fluorescent protein (mRFP) in pCS2 +was a gift from Dr. N. Kinoshita (National Institute for Basic Biology, Okazaki, JPN; *Iioka et al., 2004*). Palmitoylated membrane-binding green fluorescent protein (mGFP) in pCS2 +was a gift from Dr. R.M. Harland (UC Berkeley, California, USA). Each membrane label was injected at 200 pg/blastomere. Lifeact-GFP, an F-actin binding peptide attached to green fluorescent protein, was a gift from Dr. C.P. Heisenberg (Institute of Science and Technology Austria, Klosterneuburg, AUT; *Riedl et al., 2008*). Lifeact-Ruby was a gift from Dr. M. Tada (University College London, London, UK). Lifeact was injected at 40 pg/blastomere. GFP-fused alpha-catenin ($\alpha$-catenin-GFP) was a gift from Dr. W.J. Nelson (Stanford University, California, USA; *Schepis et al., 2012*), and injected at 200 pg/blastomere. Rab5c-CFP was a gift from Dr. M. Brand (Center for Regenerative Therapies, Dresden, GER; *Yu et al., 2009*), and injected at 500 pg/blastomere. Full-length ephrinB1 (eB1FL) was from Dr. I.O. Daar (Center for Cancer Research, USA; *Jones et al., 1998*), and was tagged by sub-cloning eB1FL into pCS2 +8 CmCherry (eB1-mCh; Addgene, Cambridge, MA). For localisation, eB1-mCh was injected at 200 pg/blastomere. For overexpression, eB1FL was injected at 900 pg/blastomere. Capped messenger RNA was synthesised from linearised constructs using mMessage mMachine (Ambion) as per manufacturer instructions. To assist with RNA isolation, GlycoBlue (50 µg/mL; Ambion, Austin, TX) was added to the reaction during isopropanol precipitation. RNA was prepared in Gurdon's Buffer (15 mM Tris, 88 mM NaCl, 1 mM KCl, pH 7.5) for microinjection.

### Morpholino Oligonucleotides (MO)

MO (GeneTools, Philomath, OR) were reconstituted in autoclaved double-distilled water. Four-cell stage embryos were injected in the vegetal hemisphere to target the prospective endoderm. C-cadherin MO: 5′-CCACCGTCCCGAACAGAAGCCTCAT-3′ (CcadMO) was previously characterised (*Ninomiya et al., 2012*), and injected at 10 ng/blastomere. The ephrinB1 MO: 5′-GGAGCCCTTCCATCCGCACAGGTGG-3′ (eB1MO) was previously characterised (*Rohani et al., 2011*), and injected at 10 ng/blastomere. Standard control MO: 5′-CCTCTTACCTCAGTTACAATTTATA-3′, was injected at 10 ng/blastomere.

### Chemicals

To mark the interstitial space, fluorescein isothiocyanate conjugated avidin (AvidinFITC; Sigma-Aldrich) was used (1:500 dilution) in 1X MBS. To monitor uptake of vesicles, fluorescein conjugated dextrans (10 kDa, Molecular Probes, Eugene, OR) were added to 1X MBS at 5 mg/mL. For competitive inhibition of fibronectin (FN) interaction, a synthetic hexapeptide homologous to the cell binding motif of FN: H-Gly-Arg-Gly-Asp-Ser-Pro-OH (GRGDSP; Calbiochem, San Diego, CA), and the corresponding control peptide H-Gly-Arg-Gly-Glu-Ser-Pro-OH (GRGESP; Calbiochem) were used. Peptides were reconstituted in 1X MBS (50 mg/mL).

## Microsurgery

Microsurgical manipulations were carried out on sterilised plasticine-coated petri dishes in 1X MBS. Tissues were excised with eyebrow knife and hair-loop tools then transferred onto glass-bottom dishes (MatTek, Ashland, MA) or tissue-culture dishes (Cellstar, Germany) rendered non-adhesive by pre-coating with 1% bovine serum albumin (w/v, BSA) solution in 1X MBS. Dissections were carried out under a Zeiss Stemi SV 11 microscope, observed with a Zeiss AxioCam MRc digital camera using AxioVision 4.8 software. Vegetal explants were excised from stage 10 gastrulae. A mid-sagittal

vegetal slice (without the blastocoel roof) about 5 cell layers thick was secured onto a glass bottom dish with a cover glass, altogether held in place by silicon grease as described (*Winklbauer, 1998*).

## Fluorescence microscopy

All live recordings were captured in an ambient temperature of 21–23°C. Bright-field and epi-fluorescence time-lapse videos were recorded on a Zeiss Axiovert 200 M inverted microscope with Zeiss AxioCam MRm digital camera using AxioVision 4.8 software. High resolution images were captured with a Leica TCS SP8 confocal laser scanning microscope equipped with HCX-PL-APO-CS 10x/ NA0.40, HC-PL-APO-CS 20x/NA0.75, 40x/NA1.30, 63x/NA1.20, HC-PL-APO-CS2 100x/NA1.4 oil-immersion objectives, and HCX-IRAPO-L 25x/NA0.95, HC-PL-APO-CS2 63x/NA1.20 water-immersion objectives and resonant scanning system using Leica LAS AF 3.2 software.

## Scanning electron microscopy

Embryos were fixed in 2.5% glutaraldehyde, and 2% paraformaldehyde in 50 mM HEPES buffer (pH 7.4) overnight at 4°C. Embryos were rinsed in 0.1 M sodium cacodylate (pH 7.0) and bisected mid-sagittally using a microsurgical knife. Post-fixation was performed by incubating embryos with 1% osmium tetroxide in 0.1 M sodium cacodylate overnight at 4°C. Samples were then dehydrated through a graded ethanol series (30%, 50%, 70%, 2 × 100%), dried overnight, and sputter-coated with gold–palladium. Images were obtained using the Hitachi S2500 scanning electron microscope.

## Transmission electron microscopy

X. *laevis* gastrulae with the vitelline membrane removed were gently punctured in the blastocoel roof with a tungsten needle to facilitate stain infusion. Perforated gastrulas were fixed for one week at 4°C in 3% glutaraldehyde (Fisher Scientific, Pittsburgh, PA), 2% paraformaldehyde (Fisher Scientific), and 1% alcian blue (Sigma-Aldrich) in 0.05 M sodium cacodylate (Sigma-Aldrich, pH 7.0). Embryos were rinsed in 0.1 M sodium cacodylate (pH 7.0), bisected mid-sagittally and embedded in 3% low-melting point agarose (Sigma-Aldrich), then fixed overnight at 4°C in 1% osmium tetroxide (Electron Microscopy Sciences, Hatfield, PA), and 1% lanthanum nitrate (Sigma-Aldrich) in 0.1 M sodium cacodylate (pH 7.0). Embryos were then rinsed in 0.1 M sodium cacodylate (pH 7.0) and dehydrated in a graded series of ethanol solutions (30%, 50%, 70%, 90%, 2 × 100%) then gradually infiltrated with Spurr's resin overnight at room temperature. Embryos were then cured at 65°C overnight. Semi-thin (1–1.5 μm) sections were made using a Leica RM2235 microtome, and ultrathin (90–100 nm) sections were prepared using a Leica EM UC6 microtome. Semi-thin sections were stained with 1% toluidine blue for sample inspection under bright-field microscopy, while ultrathin sections were stained using 3% uranyl acetate and Reynold's lead citrate to provide contrast for imaging using the Hitachi HT7700 transmission electron microscope operated at 80 kV.

## Tissue surface tension assay

Tissue aggregates round up into drop-shapes *in vitro*. The drop-shape of an aggregate at equilibrium represents a balance of forces between tissue surface tension (which acts to minimise the tissue into a sphere) and gravity (which acts to flatten the tissue). Given that gravity is known, tissue surface tension can be deduced by the Laplace equation using the curvature radii of the aggregate profile (*David et al., 2009*). Since surface tension is numerically equal to surface energy, which is the energy required to expand the surface of a droplet by a unit of area which is also equal to half of the energy required to split a droplet into two equal parts (i.e. creating two new surfaces); tissue surface tension is an effective measure of tissue cohesion. Tissue excised from the early gastrula (at NF stage 10 unless otherwise specified) were placed in a well for 30 min to facilitate rounding. Tissue aggregates were transferred onto BSA-coated tissue culture dishes to equilibrate for 2 hr before aggregate curvature profiles were imaged using an inclined mirror calibrated at a 45° angle to the substrate surface. Tissue surface tension was quantified using a modified Axisymmetric Drop-Shape Analysis (ADSA; Del *Río and Neumann, 1997*) adapted for use with tissue explants (*David et al., 2009*; *Luu et al., 2011*; *David et al., 2014*).

## Single cell migration assay

Endoderm cells were isolated using tissue dissected from a central column of the vegetal cell mass between the vegetal pole and the blastocoel floor and incubated in cell dissociation buffer (88 mM NaCl, 1 mM KCl, 2.4 mM NaHCO$_3$, 10 mM Hepes (+NaOH), pH 7.4) to separate cells. Culture dishes were coated with 10% (w/v) fish skin gelatin (Sigma-Aldrich) over-night then surface coated with bovine plasma FN (5 µg/cm$^2$; Sigma-Aldrich) for 2 hr at room-temperature prior to seeding. Cells were then seeded onto FN–gelatin dishes and incubated in modified Holtfreter's Solution (59 mM NaCl, 0.67 mM KCl, and 2.4 mM NaHCO$_3$).

## Morphological metrics

The cell body length–width ratio (LWR) provides a measure of cell elongation parallel to the long axis of the cell by dividing the length of the cell by its width perpendicular to the long axis. LWR was measured using cell outlines captured from time-lapse videos of membrane-labelled explants and scanning electron micrographs of mid-sagittally fractured embryos in various gastrulation stages as indicated. To assess whether cell elongation was congruent with the direction of cell migration, we analysed cell congruity ratio with respect to the animal-vegetal trajectory. Cell body orientation was measured by determining the angle of the long-axis of cells relative to the vertical axis. To determine the long-axis of individual endoderm cells, best-fit ellipses were matched to traced cell outlines (*Blanchard et al., 2009*). Individual fittings were manually checked to contain the same area, orientation and centroid as the original cell. The long-axis of the ellipse was then used as to represent the long-axis of cells. Cell movement trajectories were tracked using MtrackJ (*Meijering et al., 2012*). Cell displacement and velocity measurements were made by connecting the centroid of individual cells from lapsed time-points. Cell path alignment was measured by determining the angle of cell displacement path relative to the long-axis of cells.

## Statistical analysis

All experiments were replicated at least three times, and representative images are shown. Statistical testing was conducted using two-tailed unpaired Student's *t*-tests to compare different populations. Significant findings are shown by asterisks indicating p-values<0.05 (*),<0.01 (**), and <0.001 (***). Standard deviation (S.D.) or standard error of the mean (S.E.) are indicated. Figures were cropped and presented after intensity adjustment using Photoshop (Adobe Systems, San Jose, CA). All adjustments were performed equally within each experiment. Figures were composed using Illustrator (Adobe Systems).

# Acknowledgements

We thank Dr. AEE Bruce, Dr. U Tepass and SJ Mason for critical reading of the manuscript, Dr. SE Lepage and O Luu for technical assistance, and all members of the RW lab for discussion and comments.

# Additional information

### Funding

| Funder | Grant reference number | Author |
|---|---|---|
| Canadian Institutes of Health Research | MOP-53075 | Rudolf Winklbauer |

The funders had no role in study design, data collection and interpretation, or the decision to submit the work for publication.

### Author contributions

Jason WH Wen, Conceptualization, Formal analysis, Investigation, Writing—original draft, Writing—review and editing; Rudolf Winklbauer, Conceptualization, Resources, Supervision, Funding acquisition, Investigation, Writing—original draft, Project administration, Writing—review and editing

## Author ORCIDs

Jason WH Wen https://orcid.org/0000-0001-7402-5073
Rudolf Winklbauer http://orcid.org/0000-0002-0628-0897

## Ethics

Animal experimentation: Research animals were used in accordance with guidelines approved by the University Animal Care Committee (Protocol no. 20011765, University of Toronto, Canada).

## Decision letter and Author response

Decision letter https://doi.org/10.7554/eLife.27190.040
Author response https://doi.org/10.7554/eLife.27190.041

## Additional files

### Supplementary files

• Transparent reporting form
DOI: https://doi.org/10.7554/eLife.27190.038

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
