## [Decision Letter]

Thank you for submitting your article "Ingression-type cell migration drives vegetal endoderm internalisation in the *Xenopus* gastrula" for consideration by *eLife*. Your article has been reviewed by three peer reviewers, one of whom is a member of our Board of Reviewing Editors, and the evaluation has been overseen by Diethard Tautz as the Senior Editor. The reviewers have opted to remain anonymous.

The reviewers have discussed the reviews with one another and the Reviewing Editor has drafted this decision to help you prepare a revised submission.

Summary:

Although vegetal internalization movements were observed in amphibian gastrulae as long ago as the 1930's, it was only in 1999 that Dr. Winklbauer and colleagues described in detail the extensive reshaping of the vegetal endoderm in gastrulating *Xenopus* embryos and explants and recognized that it ("vegetal rotation" as they called it) is an active, region-specific, morphogenetic process essential to *Xenopus* gastrulation. After an interval of some years, the current article by Drs. Winklbauer and Wen provides new information about the patterned cell rearrangements occurring during vegetal internalization as a whole and on the cellular activities that underlie internalization. Results include:

1) Characterization (with morphometrics) in the whole embryo and explants of the structured heterogeneity of cell behaviors in the animal-to-vegetal and central-to-peripheral dimensions of the vegetal endoderm with regard to speeds of movement of the large yolky endoderm cells, the size of intercellular spaces, the contacts between cells, and the changes of direction as cells approach and enter the blastocoel floor, greatly expanding its surface area (more than 20-fold as indicated by the red lines in Figure 1) and apposing the gastrulated mesoderm and ectoderm layers of the embryo. Deep endodermal cells appear to undergo extensive, relative migration, often moving in past each other in a vegetal to animal direction, perhaps with temporarily immobile neighbors as a substratum for intermittent moving cells, which in turn stop moving and serve as substratum. The authors suggest that this "differential migration" of cells is largely responsible for the reshaping of the vegetal endoderm.

2) Characterization at the individual cell level of changes of cell shape and neighbor contacts during morphogenesis, by use of a combination of morphometrics, TEM, SEM, fluorescence microscopy (e.g., membrane and actin labeling), and movies of live cells in explants and on gelatin-fibronectin surfaces. Cell movement seems to occur in the absence of forward-projecting lamellipodia or other protrusions often seen in migrating cells, and instead to involve a form of amoeboid locomotion, which the authors characterize. They identify ephrinB1-dependent macropinocytosis and *trans*-endocytosis, and active filopodial protrusive activity at the trailing edges of cells, and transient c-cadherin-dependent filiform protrusions along the cells' lateral surfaces. They suggest that amoeboid movement involves a cyclic bulging out of the anterior end of the cell, anterior-directed movement of cytoplasm, and recession and retraction of the tail end with a trafficking of vesicles during retraction (macropinocytosis and transcytosis). This is the first characterization of cell behaviors for this kind of morphogenesis. They report the cessation of vegetal internalization movements after knockdown and overexpression of fibronectin (matrix), C-cadherin (cell contacts and signaling), and ephrin B1 (cell-cell signaling). The Winklbauer lab has used many of these methods in previous published studies. The 8 movies are helpful as some of the key observations are best shown in them.

3) The authors suggest that "ingression-type migration" is a term best describing the morphogenetic activity of vegetal endoderm cells, after they compare it to the ingressions and bottle cell invaginations preformed by other cell populations during gastrulation. They note that ingression-type migration probably arose in the evolution of fish (except teleosts) and amphibians with large yolky eggs, as a means to deal with a yolky multilayered endoderm portion of the embryo, whereas gastrulating embryos of most other animal groups have only a single-layered vegetal region that internalizes (e.g. ascidian, amphioxus, echinoderms, *Drosophila*).

Essential revisions:

The three reviewers found the work to be a novel and valuable contribution to the area of cellular/developmental biology, specifically, mechanisms of morphogenesis. The authors are the experts on vegetal internalization-Dr. Winklbauer discovered it. The manuscript is well organized and clearly written (see minor comments below for further clarification). The paper is challenging to read closely since there are 11 figures, many with 8-10 panels, and then 8 movies, with many images and morphometric analyses to scrutinize and to integrate with written text and legends. Nonetheless, the manuscript form seems suitable for the author's thorough presentation of complex visual material and morphometric analysis. The reviewers expect that after the authors attend to various issues of the written presentation, the manuscript will likely be acceptable for publication.

The authors have three main conclusions about vegetal internalization to get across to the reader, namely, the existence and importance of:

1) the patterned "differential migration" of cells within the vegetal endoderm,

2) the amoeboid movement cycle of cells as driving the deformation of the vegetal endoderm, and

3) ingression-type migration as a term best characterizing the overall morphogenesis.

The authors provide clear summary statements of the conclusions, but we suggest that they indicate through their wording that these are models and interpretations seeming to them to best fit the data, as opposed to final proofs.

Related to this, the authors have focused mostly on the migration aspect of vegetal endoderm internalization. However, as they introduce the Results section, they say, "…this tissue deformation must be due to cell rearrangement, cell shape change, cell orientation, or any combination of these factors." And, "…our data support the notion that a combination of cell rearrangement and oriented cell elongation underlie the distinct shape change of vegetal explants." Then the results focus mostly on amoeboid migration-driven, cell-on-cell, rearrangements as the major contributor to overall morphogenesis.

We ask the authors to give the reader a roughly quantitative evaluation of relative contributions of other factors to the overall morphogenesis such as cell shape change (a lot of new surface is generated, vegetal central cells seem to get long and thin), and intercalation of cells in the z-axis (see the various movies in which the z-axis is the mediolateral axis of the embryo) that also deserve consideration. We don't think new data, such as additional movies are needed, but a rough quantification of the relative contributions other processes would help the reader to assess the magnitude of contribution of amoeboid migration to the overall morphogenesis.

Reviewers ask if there might be a better term that "vegetal rotation"-such as just vegetal internalization or vegetal ingression-since "rotation" doesn't seem to describe much of the morphogenesis.

---

## [Author Response]

*Essential revisions:*

*The three reviewers found the work to be a novel and valuable contribution to the area of cellular/developmental biology, specifically, mechanisms of morphogenesis. The authors are the experts on vegetal internalization-Dr. Winklbauer discovered it. The manuscript is well organized and clearly written (see minor comments below for further clarification). The paper is challenging to read closely since there are 11 figures, many with 8-10 panels, and then 8 movies, with many images and morphometric analyses to scrutinize and to integrate with written text and legends. Nonetheless, the manuscript form seems suitable for the author's thorough presentation of complex visual material and morphometric analysis. The reviewers expect that after the authors attend to various issues of the written presentation, the manuscript will likely be acceptable for publication.*

*The authors have three main conclusions about vegetal internalization to get across to the reader, namely, the existence and importance of:*

*1) the patterned "differential migration" of cells within the vegetal endoderm,*

*2) the amoeboid movement cycle of cells as driving the deformation of the vegetal endoderm, and*

*3) ingression-type migration as a term best characterizing the overall morphogenesis.*

*The authors provide clear summary statements of the conclusions, but we suggest that they indicate through their wording that these are models and interpretations seeming to them to best fit the data, as opposed to final proofs.*

We rephrased the summary to better distinguish between observations and interpretations.

*Related to this, the authors have focused mostly on the migration aspect of vegetal endoderm internalization. However, as they introduce the Results section, they say, "…this tissue deformation must be due to cell rearrangement, cell shape change, cell orientation, or any combination of these factors." And, "…our data support the notion that a combination of cell rearrangement and oriented cell elongation underlie the distinct shape change of vegetal explants." Then the results focus mostly on amoeboid migration-driven, cell-on-cell, rearrangements as the major contributor to overall morphogenesis.*

*We ask the authors to give the reader a roughly quantitative evaluation of relative contributions of other factors to the overall morphogenesis such as cell shape change (a lot of new surface is generated, vegetal central cells seem to get long and thin), and intercalation of cells in the z-axis (see the various movies in which the z-axis is the mediolateral axis of the embryo) that also deserve consideration. We don't think new data, such as additional movies are needed, but a rough quantification of the relative contributions other processes would help the reader to assess the magnitude of contribution of amoeboid migration to the overall morphogenesis.*

We had indeed neglected to consider appropriately the movement of cells along the z-axis, and had not sufficiently discussed the relative contributions of the various processes. We addressed these points by adding two and a half paragraphs to the first Results section, a new panel to Figure 2 (i.e. new Figure 2), and multiple lines where appropriate in the following sections. Overall, we confirmed that cell rearrangement by migration is the main driver of endoderm tissue shape change. Cell elongation in an orientated fashion contributes to a lesser extent, but it is part of the amoeboid mode of cell migration anyway and thus not an independent mechanism. However, as the reviewers pointed out, intercalation in the z-plane was significant, and we took this process into consideration. In the time lapse recordings, we found no clear evidence that cells were actively migrating from deep layers into the surface layer of explants. Instead, it appeared that surface cells were moving apart locally to expose deep cells below, in this way giving rise to a z-plane intercalation event. We therefore assumed that migration occurs mainly in the x-y-plane of the explant, and z-plane intercalations are mostly “passive”. We acknowledge, however, that near the blastocoel floor a z-component of active movement would actually be expected, due to cell re-orientation.

*Reviewers ask if there might be a better term that "vegetal rotation"-such as just vegetal internalization or vegetal ingression-since "rotation" doesn't seem to describe much of the morphogenesis.*

The term “vegetal rotation” was introduced with the original description of the process (Winklbauer and Schürfeld, 1999), and used since then by us and others for lack of a better term, although a rotational movement is indeed observed in large explants filmed over a sufficiently long time period (Winklbauer and Schürfeld, 1999), and in intact embryos using X-ray diffraction imaging (e.g. Moosmann et al., 2013) or NMR (Papan et al., 2007). With the present manuscript, we intend to replace the term on the basis of a more refined description of the process, but to refer to it, we still had to use this established term. Nevertheless, we changed the wording now so as to minimize its occurrence in the manuscript.